


**The UK contribution to CMIP6/PMIP4: mid-Holocene and Last**
**Interglacial experiments with HadGEM3, and comparison to the pre-**
**industrial era and proxy data**
**Charles J. R. Williams[1,5], Maria-Vittoria Guarino[2], Emilie Capron[3], Irene Malmierca-**
**Vallet[1,2], Joy S. Singarayer[4,1], Louise C. Sime[2], Daniel J. Lunt[1] Paul J. Valdes[1]**
[1]School of Geographical Sciences, University of Bristol, UK (c.j.r.williams@bristol.ac.uk)
[2]British Antarctic Survey, Cambridge, UK
[3]Physics of Ice, Climate and Earth, Niels Bohr Institute, University of Copenhagen, Denmark
[4]Department of Meteorology & School of Archaeology, Geography and Environmental
Science, University of Reading, UK
[5]NCAS-Climate / Department of Meteorology, University of Reading, UK
**Corresponding author address:**
Room 1.2n, School of Geographical Sciences,
University Road, Bristol, BS8 1SS
United Kingdom
Email: c.j.r.williams@bristol.ac.uk
Short title: mid-Holocene and Last Interglacial experiments with HadGEM3
Keywords: Palaeoclimate, Quaternary change, mid-Holocene, Last Interglacial





## ABSTRACT

Palaeoclimate model simulations are an important tool to improve our understanding of the mechanisms of climate change. These simulations also provide tests of the ability of models to simulate climates very different to today. Here we present the results from two simulations using the latest version of the UK's physical climate model, HadGEM3-GC3.1; the mid-Holocene (~6 ka) and Last Interglacial (~127 ka) simulations, both conducted under the auspices of CMIP6/PMIP4. These periods are of particular interest to PMIP4 because they represent the two most recent warm periods in Earth history, where atmospheric concentration of greenhouse gases and continental configuration is similar to the pre-industrial period but where there were significant changes to the Earth's orbital configuration, resulting in a very different seasonal cycle of radiative forcing.

Results for these simulations are assessed against proxy data, previous versions of the UK model, and models from the previous CMIP5 exercise. When the current version is compared to the previous generation of the UK model, the most recent version suggests limited improvement. In common with these previous model versions, the simulations reproduce global land and ocean temperatures (both surface and at 1.5 m) and a West African monsoon that is consistent with the latitudinal and seasonal distribution of insolation. The Last Interglacial simulation appears to accurately capture Northern Hemisphere temperature changes, but without the addition of Last Interglacial meltwater forcing cannot capture the magnitude of Southern Hemisphere changes. Model-data comparisons indicate that some geographical regions, and some seasons, produce better matches to the palaeodata (relative to pre-industrial) than others. Model-model comparisons, relative to previous generations same model and other models, indicate similarity between generations in terms of both the intensity and northward enhancement of the mid-Holocene West African monsoon, both of which are underestimated. On the 'Saharan greening' which occurred the mid-Holocene African Humid Period, simulation results are likewise consistent with other models. The most recent version of the UK model appears to still be unable to reproduce the amount of rainfall necessary to support grassland across the Sahara.





## 1. INTRODUCTION

Simulating past climates has been instrumental in improving our understanding of the mechanisms of climate change (e.g. Gates 1976, Haywood *et al*. 2016, Jungclaus *et al*. 2017, Kageyama *et al*. 2017, Kageyama *et al*. 2018, Kohfeld *et al*. 2013, Lunt *et al*. 2008, Otto-Bliesner *et al*. 2017, Ramstein *et al*. 1997), as well as in identifying and assessing discrepancies in palaeoclimate reconstructions (e.g. Rind & Peteet 1985). Palaeoclimate scenarios can also provide tests of the ability of models to simulate climates that are very different to today, often termed 'out-of-sample' tests. This notion underpins the idea that robust simulations of past climates improve our confidence in future climate change projections (Braconnot *et al*. 2011, Harrison *et al*. 2014, Taylor *et al*. 2011). Palaeoclimate scenarios have also been used to provide additional tuning targets for models (e.g. Gregoire *et al*. 2011), in combination with historical or pre-industrial conditions.

The international Climate Model Intercomparison Project (CMIP) and the Palaeoclimate Model Intercomparison Project (PMIP) have spearheaded the coordination of the international palaeoclimate modelling community to run key scenarios with multiple models, perform data syntheses, and undertake model-data comparisons since their initiation twenty-five years ago (Joussaume & Taylor 1995). Now in its fourth incarnation, PMIP4 (part of the sixth phase of CMIP, CMIP6), it includes a larger set of models than previously, and more palaeoclimate scenarios and experiments covering the Quaternary (documented in Jungclaus *et al*. 2017, Kageyama *et al*. 2017, Kageyama *et al*. 2018 and Otto-Bliesner *et al*. 2017) and Pliocene (documented in Haywood *et al*. 2016).

PMIP4 specifies experiment set-ups for two warm interglacial simulations: the mid-Holocene (MH) at ~6 ka and the Last Interglacial (LIG) covering ~129-116 ka. These are the two most recent warm periods in Earth history, and are of particular interest to PMIP4; indeed, the MH experiment is one of the two entry cards into PMIP (Otto-Bliesner *et al*. 2017). This is because whilst the atmospheric concentration of greenhouse gases, the extent of land ice, and the continental configuration is similar in these PMIP4 set-ups compared to the pre-industrial (PI) period, significant changes to the seasonal cycle of radiative forcing, relative to today, do occur during these periods due to long-term variations in the Earth's orbital configuration. The MH and LIG both have higher boreal summer insolation and lower boreal winter insolation compared to the PI, as shown by Figure 1, leading to an enhanced seasonal cycle in insolation as well as a change in its latitudinal distribution. The change is more significant in the LIG than the MH, due to the larger eccentricity of the Earth's orbit at that time.

Palaeodata syntheses indicate globally warmer surface conditions of potentially ~0.7°C than PI in the MH (Marcott *et al*. 2013) and up to ~1.3°C in the LIG (Fischer *et al*. 2018). Recent palaeodata compilations (Capron *et al*. 2014, Hoffman *et al*. 2017) reveal that the maximum temperatures were



reached asynchronously in the LIG between the Northern and Southern Hemispheres.  Furthermore,
model simulations suggest that this may have been caused by meltwater induced shutdown of the
Atlantic Meridional Overturning Circulation (AMOC) in the early part of the LIG, due to the melting
of the Northern Hemisphere ice sheets during the preceding deglaciation (e.g. Stone *et al*. 2016).
During both warm periods there is abundant palaeodata evidence indicating enhancement of Northern
Hemisphere summer monsoons (e.g. Wang *et al*. 2008) and in the case of the Sahara, replacement of
desert by shrubs and steppe vegetation (e.g. Drake *et al*. 2011, Hoelzmann *et al*. 1998) and inland
water bodies (e.g. Drake *et al*. 2011, Lezine *et al*. 2011).

The driving mechanism producing the climate and environmental changes indicated by the palaeodata
for the LIG and MH is different to current and future anthropogenic warming, as the former results
from orbital forcing changes whilst the latter results from increases in greenhouse gases.  However,
these past warm intervals are a unique opportunity to understand the magnitudes of forcings and
feedbacks in the climate system that produce warm interglacial conditions, which can help us
understand and constrain future climate projections (e.g. Holloway *et al*. 2016, Rachmayani *et al*.
2017, Schmidt *et al*. 2014).  Running the same model scenarios with ever newer models enables the
testing of whether model developments are producing improvements in palaeo model-data
comparisons, assuming appropriate boundary conditions are used.  Previous iterations of PMIP, with
older versions of the PMIP4 models, have uncovered persistent shortcomings (Harrison *et al*. 2015)
that have not been eliminated despite developments in resolution, model physics, and addition of
further Earth system components.  One key example of this is the continued underestimation of the
increase in rainfall over the Sahara in the MH PMIP simulations (e.g. Braconnot *et al*. 2012).

In this study we run and assess the latest version of the UK's physical climate model, HadGEM3-
GC3.1.  In Global Coupled (GC) version 3 (and therefore the following GC3.1), there have been
many updates and improvements, relative to its predecessors, which are discussed extensively in
Williams *et al*. (2017) and a number of companion scientific model development papers (see Section
2.1).  As a brief introduction, however, GC3 includes a new aerosol scheme, multilayer snow scheme,
multilayer sea ice and several other parametrization changes, including a set relating to cloud and
radiation, as well as a revision to the numerics of convection (Williams *et al*. 2017).  In addition, the
ocean component of GC3 has other changes including a new ocean and sea ice model, a new cloud
scheme, and further revisions to all parametrization schemes (Williams *et al*. 2017).  See Section 2.1
for further details.

Following the CMIP6/PMIP4 protocol, here the PMIP4 MH and LIG simulations have been
conducted and assessed, comparing the results with available proxy data, previous versions of the
UK's same physical climate model, and other models from CMIP5.  The focus of this paper is on the



fidelity of the temperature anomalies globally and the degree of precipitation enhancement in the
Sahara, the latter of which has proved problematic for several generations of models.  The results
discussed here are split into two sections: after an assessment of the level of equilibrium gained
during the spin-up phase, the main focus is on the model-data and model-model comparisons using
the production runs.  Following this introduction, Section 2 describes the model, the experimental
design and the proxy data used for the model-data comparisons.  Section 3 then presents the results,
divided into two subsections: i) equilibrium during the spin-up phase; and ii) model-data and model-
model comparisons from the production runs.  Finally, section 4 summarises and concludes.

**2. MODEL, EXPERIMENT DESIGN AND DATA**
**2.1. Model**
The MH and LIG simulations conducted here (referred to as *midHolocene* and *lig127k*, respectively,
and collectively as the 'warm climate' simulations), and indeed the PI simulation (*piControl*,
conducted elsewhere as part of the UK's CMIP6 runs and used here for comparative purposes) were
all run using the same fully-coupled GCM: the Global Coupled 3 configuration of the UK's physical
climate model, HadGEM3-GC3.1.  Full details on HadGEM3-GC3.1, and a comparison to previous
configurations, are given in Williams *et al.* (2017) and Kuhlbrodt *et al.* (2018).  Here, the model was
run using the Unified Model (UM), version 10.7, and including the following components: i) Global
Atmosphere (GA) version 7.1, with an N96 atmospheric spatial resolution (approximately 1.875°
longitude by 1.25° latitude) and 85 vertical levels; ii) the NEMO ocean component, version 3.6,
including Global Ocean (GO) version 6.0 (ORCA1), with an isotropic Mercator grid which, despite
varying in both meridional and zonal directions, has an approximate spatial resolution of 1° by 1° and
75 vertical levels; iii) the Global Sea Ice (GIS) component, version 8.0 (GSI8.0); iv) the Global Land
(GL) configuration, version 7.0, of the Joint UK Land Environment Simulator (JULES); and v) the
OASIS3 MCT coupler.  The official title for this configuration of HadGEM3-GC3.1 is HadGEM3-
GC31-LL N96ORCA1 UM10.7 NEMO3.6 (for brevity, hereafter HadGEM3).

All of the above individual components are summarised by Williams *et al.* (2017) and detailed
individually by a suite of companion papers (see Walters *et al.* 2017 for GA7 and GL7, Storkey *et al.*
2017 for GO6 and Ridley *et al.* 2017 for GIS8).  However, a brief description of the major changes
relative to its predecessor are given here.  Beginning with GA7 and GL7, a once-in-a-decade
replacement of the model's dynamical core, implementing ENDGame, was undertaken for the
previous version (GA6) and therefore remains the same in GA7 (Walters *et al.* 2017).  In addition, a
number of bottom-up and top-down developments were included in GA7.  For the former, these
include improvements to the radiation scheme to allow better treatment of gases absorption,
improvements to how warm rain and ice clouds are treated, and an improvement to the numerics of
the convection scheme (Walters *et al.* 2017).  For the latter, these include further improvements to the
microphysics as well as an incremental development of ENDGame (Walters *et al*. 2017).  Together
these led to reductions in four model errors that were deemed critical in the previous configuration: i)
South Asian monsoon rainfall biases over India; ii) biases in both temperature and humidity in the
tropical tropopause; iii) shortcomings in the numerical conservation; and iv) biases in surface
radiation fluxes over the Southern Ocean (Walters *et al*. 2017).  In addition to these developments,
two new parameterisation schemes were introduced in GA7: firstly the UK Chemistry and Aerosol
(UKCA) GLOMAP-mode aerosol scheme, to improve the representation of tropospheric aerosols, and
secondly a multi-layer snow scheme in JULES, to allow the first time inclusion of stochastic physics
in UM climate simulations (Walters *et al*. 2017).

For the GO and GIS components, a number of improvements to GO6 have been made since the
previous version, the first of which was an upgrade of the NEMO base code (to version 3.6) which
allowed a formulation for momentum advection (from Hollingsworth *et al*. 1983), a Lagrangian
icebergs scheme, and a simulation of circulation below ice shelves (Storkey *et al*. 2018).  Other
developments included an improvement to the warm SST bias in the Southern Ocean (as detailed by
Williams *et al*. 2017), as well as tuning to various parameters e.g. the isopycnal diffusion (Storkey *et*
*al*. 2018).  For GIS8, along with improvements to the albedo scheme and more realistic semi-implicit
coupling, the biggest development since its predecessor is the inclusion of multilayer
thermodynamics, giving a heat capacity to the sea ice and allowing vertical variation of conduction
(Ridley *et al*. 2018).  Testing of these two components produced a better simulation compared to its
predecessor, with more realistic mixed layer depths in the Southern Ocean and the aforementioned
reduced warm bias, the latter of which was deemed primarily due to the tuning of the different mixing
(e.g. vertical and isopycnal) parameters (Storkey *et al*. 2018).

When all of these components are coupled together to give GC3, there have been several
improvements relative to its predecessor (GC2), most noticeably to the large warm bias in the
Southern Ocean (which was reduced by 75%), as well as an improved simulation of clouds, sea ice,
the frequency of tropical cyclones in the Northern Hemisphere as well as the AMOC, and the Madden
Julian Oscillation (MJO) (Williams *et al*. 2017).  Relative to the previous fully-coupled version of the
model (HadGEM2), which was submitted to the last CMIP5/PMIP3 exercise, many systematic errors
have been improved including a reduction in many regions to the temperature bias, a better simulation
of mid-latitude synoptic variability, and an improved simulation of tropical cyclones and the El Niño
Southern Oscillation (ENSO) (Williams *et al*. 2017).

Here, the *midHolocene* and *lig127k* simulations were both run on the UK National Supercomputing
Service, ARCHER, whereas the *piControl* was run on a different platform based within the UK Met
Office's Hadley Centre.  While this may mean that anomalies computed against the *piControl* are

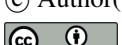



potentially influenced by different computing environments, and not purely the result of different
climate forcings, the reproducibility of GC3.1 simulations across different platforms has been tested
(Guarino *et al*. 2019). It was found that, although a simulation length of 200 years is recommended
whenever possible to adequately capture climate variability across different platforms, the main
climate variables considered here (e.g. surface temperature) are not expected to be significantly
different on a 100- or 50-year timescale (see, for example, Fig. 6 in Guarino *et al*. [2019]) as they are
not directly affected by medium-frequency climate processes such as ENSO.

Not including queueing time, both simulations were achieving 3-4 model years per day during the
spin-up phase, and 1-2 model years per day during the production run; see below for the differences in
output, and therefore speed, between the two phases.

### 2.2. Experiment design

Full details of the experimental design, and results from the CMIP6 *piControl* simulation, are
documented in Menary *et al*. (2018). Both the warm climate simulations followed the experimental
design given by Otto-Bliesner *et al*. (2017), and specified at
https://pmip4.lsce.ipsl.fr/doku.php/exp_design:index. The primary differences from the *piControl*
were to the astronomical parameters and the atmospheric trace greenhouse gas concentrations,
summarised in Table 1. For the astronomical parameters, these were prescribed in Otto-Bliesner *et al*.
(2017) according to orbital constants from Berger & Loutre (1991). However, in HadGEM3, the
individual parameters (e.g. eccentricity, obliquity, etc) use orbital constants based on Berger (1978),
according to the specified start date of the simulation. For the atmospheric trace greenhouse gas
concentrations, these were based on recent reconstructions from a number of sources (see Table 1 for
values, and section 2.2 in Otto-Bliesner *et al*. [2017] for a full list of references/sources).

All other boundary conditions, including solar activity, ice sheets, topography and coastlines, volcanic
activity and aerosol emissions, are identical to the CMIP6 *piControl* simulation. Likewise, vegetation
was prescribed to present-day values, to again match the CMIP6 *piControl* simulation. As such, the
*piControl* and both the warm climate simulations actually include a prescribed fraction of urban land
surface. As a result of this, our orbitally- and greenhouse gas-forced simulations should be considered
as anomalies to the *piControl*, rather than absolute representations of the MH or LIG climate.

Both the warm climate simulations were started from the end of the *piControl* spin-up phase (which
ran for approximately 600 years), after which time the *piControl* was considered to be in atmospheric
and oceanic equilibrium (Menary *et al*. 2018). To assess this, four metrics were used, namely net
radiative balance at the top of the atmosphere (TOA), surface air temperature (SAT), and full-depth
ocean temperature (OceTemp) and salinity (OceSal) Menary *et al*. (2018). See Section 3.1 (and in
particular Table 2) for an analysis of the equilibrium state of both the *piControl* and the warm climate
simulations.  Starting at the end of the *piControl*, these were then run for their own spin-up phases,
400 and 350 years for the *midHolocene* and *lig127k* respectively.  During this phase, ~700 diagnostics
were output, containing mostly low temporal frequency (e.g. monthly, seasonal and annual) fields.
Once the simulations were considered in an acceptable level of equilibrium (see Section 3.1), a
production phase was run for 100 and 200 years for the *midHolocene* and *lig127k* respectively, during
which the full CMIP6/PMIP4 diagnostic profile (totalling ~1700 fields) was implemented to output
both high and low temporal frequency variables.

**2.3. Data**
Recent data syntheses compiling quantitative surface temperature and rainfall reconstructions were
used in order to evaluate the warm climate simulations.

For the MH, the global-scale continental surface mean annual temperature (MAT) and rainfall (or
mean annual precipitation, MAP) reconstructions from Bartlein *et al*. (2011), with quantitative
uncertainties accounting for climate parameter reconstruction methods, were used (see Data
Availability for access details).  They rely on a combination of existing quantitative reconstructions
based on pollen and plant macrofossils and are inferred using a variety of methods (see Bartlein *et al*.
2011 for further details).  At each site, the 6 ka anomaly (corresponding to the 5.5-6.5 ka average
value), is given relative to the present day, and in the case where modern values could not be directly
inferred from the record, modern climatology values (1961-1990) were extracted from the Climate
Research Unit historical climatology data set (New *et al*. 2002).

For the LIG, two different sets of surface temperature data are available.  Firstly, the Capron *et al*.
(2017) 127 ka timeslice of SAT and sea surface temperature (SST) anomalies (relative to pre-
industrial, 1870-1899), is based on polar ice cores and marine sediment data that are (i) located
poleward of 40° latitude and (ii) have been placed on a common temporal framework (see Data
Availability for access details).  Polar ice core water isotope data are interpreted as annual surface air
temperatures, while most marine sediment-based reconstructions are interpreted as summer SST
signals.  For each site, the 127 ka value was calculated as the average value between 126 and 128 ka
using the surface temperature curve resampled every 0.1 ka.  Secondly, a global-scale time slice of
SST anomalies, relative to pre-industrial (1870-1889), at 127 ka was built, based on the recent
compilation from Hoffman *et al*. (2017), which includes both annual and summer SST reconstructions
(see Data Availability for access details).  The 127 ka values at each site were extracted, following the
methodology they proposed for inferring their 129, 125 and 120 ka time slices i.e. the SST value at
127 ka was taken on the provided mean 0.1 ka interpolated SST curve for each core location.  Data
syntheses from both Capron *et al*. (2014, 2017) and Hoffman *et al*. (2017) are associated with



quantitative uncertainties accounting for relative dating and surface temperature reconstruction
methods. Here, the two datasets are treated as independent data benchmarks, as they use different
reference chronologies and methodologies to infer temporal surface temperature changes, and
therefore they should not be combined.  See Capron *et al*. (2017) for a detailed comparison of the two
syntheses.  A model-data comparison exercise using existing LIG data compilations focusing on
continental surface temperature (e.g. Turney and Jones 2010) was not attempted, as they do no benefit
yet from a coherent chronological framework, preventing the definition of a robust time slice
representing the 127 ka terrestrial climate conditions (Capron *et al*. 2017).

**3. RESULTS**
As briefly mentioned above, both the warm climate simulations had a spin-up phase before the main
production run was started.  The results discussed here are therefore split into two sections: firstly,
assessing the level of atmospheric and oceanic equilibrium during (and, in particular, at the end of)
the spin-up phase, and secondly assessing the 100-year climatology from the production run.

**3.1. Spin-up**
Annual global mean 1.5 m air temperature and TOA radiation from both warm climate simulations,
compared to the *piControl*, are shown in Figure 2 and summarised in Table 2.  Note that the *piControl*
spin-up phase was run in three separate parts, to accommodate for minor changes/updates in the
model as the simulation progressed.  There is a clear increase in temperature during the beginning of
this period, as the *piControl* slowly spins up from its original starting point; this levels off towards the
end of the period, however, with a final temperature trend of 0.03°C century$^{-1}$ (Table 2 and Fig. 2a).
For the warm climate simulations, despite considerable interannual variability (particularly halfway
through the *lig127k* simulation) both are showing small long-term trends of -0.06°C century$^{-1}$ and -
0.16°C century$^{-1}$ for the last 100 years of the *midHolocene* and *lig127k,* respectively (Table 2 and Fig.
2a).  The same is true for TOA, where the *piControl* has a slow downward trend towards zero until
equilibrium was reached, whereas the *midHolocene* and *lig127k* are relatively stable (Fig. 2b).

For the ocean, annual global mean OceTemp and OceSal are shown in Table 2 and Figure 3.  There is
again a clear increase in OceTemp during the *piControl* spin-up phase, which again stabilises at
0.035°C century$^{-1}$ by the end of the period (Table 2).  Whilst OceTemp stabilises in the *midHolocene*
and indeed has a smaller trend than the *piControl* (Table 2), it continues to increase in the *lig127k*
until it stabilises within the last ~50 years (Fig. 3a).  A similar pattern is shown in OceSal, with a
steady decrease in the *piControl* spin-up phase which continues during the *midHolocene* and,
conversely, starts to increase before stabilising during the *lig127k* (Fig. 3b).  Concerning the long-
term trends, Menary *et al*. (2018) considered values acceptable for equilibrium to be < +/-0.035°C
century$^{-1}$ and < +/-0.0001 psu century$^{-1}$ (for OceTemp and OceSal, respectively); as shown in Table 2,

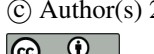



although both warm climate simulations meet the temperature criterion, neither meet the salinity
criterion (-0.007 psu and 0.006 psu for the *midHolocene* and *lig127k,* respectively, compared to a
criterion of 0.0001 psu).  However, running for several thousands of years (and > 5 years of computer
time), which would be needed to reach true oceanic equilibrium, was simply unfeasible here given
time and resource constraints.

**3.2. Production runs results**
The warm climate production runs were undertaken following the spin-up phase, with a 100-year
climatology of each simulation being compared to that from the *piControl*, as well as available proxy
data, using either annual means or summer/winter seasonal means.  For the latter, depending on the
availability of the proxy data, Northern Hemisphere summer is defined as either June-August (JJA) or
July-September (JAS), and Northern Hemisphere winter is defined as either December-February
(DJF) or January-March (JFM); and vice versa for Southern Hemisphere summer/winter.  Using
atmospheric diagnostics, the focus is on three separate measures: i) to describe and understand the
differences between the current two warm climate simulations and the *piControl* in terms of
temperature, rainfall and atmospheric circulation changes; ii) to compare both current simulations,
with existing and newly-available proxy data, and iii) to compare both current simulations with those
from previous versions of the UK model (where available), such as HadGEM2-ES or HadCM3, in
order to assess any improvements due to model advances.  In this aim, previous CMIP3 and 5
versions of the UK model, alongside other CMIP5 models, will be assessed to address the question of
whether simulations produce enough rainfall to allow vegetation growth across the Sahara: the mid-
Holocene 'Saharan greening' problem.

**3.2.1. Do the CMIP6 HadGEM3 simulations show temperature, rainfall and circulation**
**differences when compared to the pre-industrial era?**
Here we focus on mean differences between the HadGEM3 warm climate simulations and the
corresponding *piControl*.  Seasonal mean summer and winter 1.5 m air temperature anomalies
(relative to the *piControl*) from both warm climate simulations are shown in Figure 4.  During JJA,
the *midHolocene* is showing a widespread increase in temperatures of up to 2°C across the entire
Northern Hemisphere north of 30°N, more in some places e.g. Greenland (Fig. 4a), consistent with the
increased latitudinal and seasonal distribution of insolation caused by known differences in the
Earth's axial tilt (Berger & Loutre 1991, Otto-Bliesner *et al*. 2017).  The only places showing a
reduction in temperature are West and central Africa (around 10°N) and northern India; this, as
discussed below, is likely related to increased rainfall in response to a stronger summer monsoon, but
could also be due to the resulting increase in cloud cover (reflecting more insolation) or a combination
of the two.  During DJF, only the Northern Hemisphere high latitudes (north of 60°N) continue this





warming trend, with the rest of continental Africa and Asia showing a reduction in temperature (Fig.
4b). These patterns are virtually the same during the *lig127k* (Fig. 4c and d), just much more
pronounced (with temperature increases during JJA of 5°C or more); again, this is consistent with the
differences in the Earth's axial tilt, which were more extreme (and therefore Northern Hemisphere
summer experienced larger insolation changes) in the LIG relative to the MH (Berger & Loutre 1991,
Otto-Bliesner *et al.* 2017).

Mean JJA rainfall and 850mb wind anomalies (relative to the *piControl*) from both warm climate
simulations are shown in Figure 5, which zooms into Africa. In response to the increased Northern
Hemisphere summer insolation, the West African monsoon is enhanced in both simulations, with
positive (negative) rainfall anomalies across sub-Saharan Africa (eastern equatorial Atlantic)
suggesting a northward displacement of the ITCZ. This is consistent with previous work, with a
northward movement of the rainbelt being associated with increased advection of moisture into the
continent (Huag *et al.* 2001, Singarayer *et al.* 2017, Wang *et al.* 2014). This increased advection of
moisture is shown by the low-level westerlies in Figure 5, drawing in more moisture from the tropical
Atlantic, which are consistent with previous work documenting the intensified monsoon circulation
associated with a greater land-sea temperature contrast (Huag *et al.* 2001, Singarayer *et al.* 2017,
Wang *et al.* 2006). This pattern is enhanced in the *lig127k* relative to the *midHolocene*, again due to
the stronger insolation forcing in the LIG relative to the MH, and the northward displacement of the
ITCZ is more pronounced in the *lig127k* simulation (Fig. 5c). Interestingly, however, regarding very
small anomalies (i.e. < 1 mm day$^{-1}$), the *midHolocene* is showing wetter conditions further north,
throughout the Sahara and up to the Mediterranean, whereas the *lig127k* simulation has small dry
anomalies in this region (Fig. 5a and b for the *midHolocene* and *lig127k*, respectively).

The change to the intensity and the spatial pattern (e.g. latitudinal positioning and extent) of the West
African monsoon is further shown in Figure 6, which shows JJA rainfall anomalies by latitude over
West Africa from both warm climate simulations. Apart from the clear drying relative to the
*piControl* between the Equator and 5°N (which comes almost entirely from the equatorial Atlantic
region), both warm climate simulations are showing a large increase in rainfall (of around 2 and 6 mm
day$^{-1}$ for the *midHolocene* and *lig127k*, respectively) during the core monsoon region i.e. between
approximately 10-15°N. In terms of the latitudinal extent, an examination of the mean rainfall by
latitude suggests that both warm climate simulations are producing a wider monsoon region (i.e. both
North and South of the Equator), with rainfall only reducing to near zero at 20°N in these simulations
compared to approximately 16°N in the *piControl* (not shown). This is again consistent with previous
work, where various theories are compared as to the reasons behind the latitudinal changes in the
rainbelt's position, one which is a symmetric expansion during boreal summer (Singarayer &
Burrough 2015, Singarayer *et al.* 2017).




**3.2.2. Model-Data comparison: Do the CMIP6 HadGEM3 simulations reproduce the**
**'reconstructed' climate based on available proxy data?**
Here we focus on comparison with recent proxy data, focusing on surface temperature and rainfall
(drawing direct comparisons, as well as using the root mean square error (RMSE), between proxy and
simulated data, summarised in Table 4a), to see how well the current warm climate simulations are
reproducing the 'observed' approximate magnitudes and patterns of change. It is worth noting that
both simulated and proxy anomalies contain a high level of uncertainty, and in many locations the
uncertainty is often larger than the anomalies themselves (not shown). The following results should
therefore be considered with this caveat in mind.

Before the spatial patterns are compared, it is useful to assess global means (focusing on 1.5 m air
temperature, calculated both annually and during Northern and Southern Hemisphere summer, JJA
and DJF respectively) for model-model comparisons. Table 3 shows these global means, where it is
clear that when annual means are considered, the *midHolocene* simulation is actually cooler than the
*piControl*; this discrepancy with the palaeodata, which in general suggests a warmer MH relative to
PI, also exists in previous models, and is termed the 'Holocene temperature conundrum' by Lui *et al.*
(2014). The *lig127k* simulation is, however, warmer than the *piControl* simulation. Given the
seasonal distribution of insolation in these two simulations, it is expected that the largest difference to
the PI occurs during boreal summer, and indeed it does; during JJA, there is a warmer LIG and a
slightly warmer MH (1.69°C and 0.07°C, respectively). Conversely, the opposite is true during DJF.

Concerning the spatial patterns during the MH, Figure 7 shows simulated surface MAT and MAP
anomalies from the *midHolocene* simulation versus MH proxy anomalies from Bartlein *et al.* (2011),
both of which have over 600 proxy locations in total (Table 4), although mostly confined to the
Northern Hemisphere. For MAT, globally the simulation looks reasonable (RMSE = 2.45°C), and
appears to be able to reproduce the sign of temperature change for many locations, with both
simulated and proxy anomalies suggesting increases in temperature North of 30°N (Fig. 7a and b).
This is not true everywhere, such as across the Mediterranean where the simulation suggests a small
warming but the proxy data indicates cooling (Fig. 7a and b). However, regarding the magnitude of
change, the *midHolocene* simulation is underestimating the temperature increase across most of the
Northern Hemisphere, with for example increases of up to 1°C across Europe from the simulation
compared to 3-4°C increases from the proxy data (Fig. 7a and b). In the simulation, temperature
anomalies only reach these magnitudes in the Northern Hemisphere polar region (i.e. north of 70°N),
not elsewhere. A similar conclusion can be drawn from MAP (RMSE = 280 mm yr$^{-1}$), where again
the *midHolocene* simulation is correctly reproducing the sign of change across most of the Northern
Hemisphere, but in some places not the magnitude. Over the eastern US, for example, rainfall



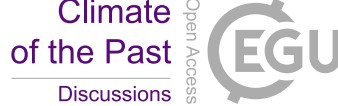

decreases of up to 200 mm yr$^{-1}$ are being shown by the simulation whereas the proxy data suggests a
much stronger drying of up to 400 mm yr$^{-1}$ (Fig. 7c and d).  Elsewhere, such as over Europe and
Northern Hemisphere Africa, the simulation more accurately reproduces the magnitude of rainfall
increases; both simulated and proxy anomalies show increases of 200-400 mm yr$^{-1}$ (Fig. 7c and d).

Concerning the spatial patterns during the LIG, Figure 8 shows simulated mean SST anomalies
(calculated both annually and during JAS/JFM) from the *lig127k* simulation and LIG proxy anomalies
from two sources, Capron *et al*. (2017) and Hoffman *et al*. (2017).  When annual anomalies are
considered, despite the lack of reconstructions in the Capron *et al*. (2017) data (Table 4), there is
relatively good agreement (RMSE = 2.44°C and 2.94°C for the Capron *et al*. (2017) and Hoffman *et*
*al*. (2017) data, respectively, and which is within the average uncertainty range), between simulated
and observed SST anomalies in the Northern Hemisphere (and in particular in the North Atlantic),
with both suggesting increased temperatures during the LIG of up to 3°C (Fig. 8a).  There are
discrepancies, such as in the Norwegian Sea, where the Hoffman *et al*. (2017) reconstructions suggest
a cooler LIG than preindustrial, whereas the *lig127k* simulation shows a consistent warming; this is,
however, consistent with previous work, and earlier climate models have also failed to capture this
cooling (Capron *et al*. 2014, Stone *et al*. 2016).  Note that, over Greenland and Antarctica, the Capron
*et al*. (2017) proxy data show SAT, not SST, and are therefore not compared in this figure;
comparison with simulated SAT, however, suggests that the model is capturing the sign, if not the
magnitude, of annual change over these regions (not shown).  During Northern Hemisphere summer,
JAS (during which period Capron *et al*. [2017] has the most proxy locations [Table 4]),, the simulated
anomalies are in agreement with many, but not all, of the proxy locations (RMSE = 3.11°C and
2.06°C for the Capron *et al*. (2017) and Hoffman *et al*. (2017) data, respectively); examples of where
they differ, not just in magnitude but also sign, again include the Norwegian and Labrador Seas (Fig.
8b).  In Southern Hemisphere summer, JFM, the model suggests a general (but weak) cooling in the
South Atlantic relative to preindustrial and a general (but weak) warming in the Southern Ocean (Fig.
8c).  However, certain proxy locations (such as off the coast of southern Africa) suggest a much
warmer LIG than preindustrial (RMSE = 1.94°C and 4.24°C for the Capron *et al*. (2017) and Hoffman
*et al*. (2017) data, respectively), which in stark contrast to the cooling in the same region from the
*lig127k* simulation (Fig. 8c) .  In the Southern Ocean, the majority of simulated anomalies reproduce
the observed sign of change, but not the magnitude; the *lig127k* simulation suggests temperature
increases of up to 1°C, whereas both proxy datasets suggest SST increases of 2-3°C depending on
location (Fig. 8c).

It would therefore be reasonable to say that, for both warm climate simulations, whilst the model is
capturing the sign and magnitude of change (for either temperature or rainfall) in some locations, this
is highly geographically dependent and there are locations where the simulation fails to capture even



the sign of change.  The model also appears to be seasonally dependent, with the *lig127k* simulation
(but not the *midHolocene* simulation) correctly reproducing both the sign and magnitude of change
during Northern Hemisphere summer in some locations, but not during Southern Hemisphere summer
or annually.

**471    3.2.3.  Model-Model comparison: Do the CMIP6 HadGEM3 simulations show an improvement**

**472    compared to older CMIP versions of the UK model?**

Here we focus on model-model intercomparisons, comparing the HadGEM3 warm climate
simulations with firstly those from previous versions of the UK model and secondly with those from
other models included in CMIP5.  It should be noted that although LIG experiments have been
conducted previously with both model-model and model-data comparisons being made (Lunt *et al*.
2013), all of these experiments were carried out using early versions of the models and were thus not
included in CMIP5.  Moreover, as part of their assessment Lunt *et al*. (2013) considered a set of four
simulations, at 130, 128, 125 and 115 ka, none of which are directly comparable to the current
HadGEM3 *lig127k* simulation.  Instead, a LIG simulation has recently been undertaken using one of
the original versions of the UK's physical climate model, HadCM3, and so this is used here to
compare with the *lig127k* simulation.  As discussed above, this section is divided into two parts:
firstly the mean climate state of the warm climate simulations will be compared to the model's
predecessors, focusing again on hydroclimate of the West African monsoon (given the known
problem of simulated rainfall underestimation in this region, see e.g. Braconnot *et al*. [2007]).  Here,
both direct comparisons and RMSE values will again be examined, this time calculating the RMSE
between the simulated rainfall anomaly from two older versions of the UK model versus the current
HadGEM3 *midHolocene* and *lig127k* simulations (summarised in Table 4b).  Secondly, previous
generation simulations (from all available models included in CMIP5) will be compared to see
whether the most recent HadGEM3 *midHolocene* simulation is now providing enough rainfall to
allow vegetation growth across the Sahara; something which previous generations of models from
CMIP5 did not (Braconnot *et al*. 2007).

***494    3.2.3.1.  Mean climate state from predecessors of HadGEM3***

Regarding the magnitude and latitudinal extent of the West African monsoon, Figure 9 shows the JJA
rainfall differences averaged over West Africa from the current *midHolocene* and *lig127k* simulation
versus two of the model's predecessors.  During the MH, the two most recent generations of the
model (HadGEM3 and HadGEM2-ES) generally agree on drier conditions over the equatorial
Atlantic and then wetter conditions over West Africa, however the oldest generation model
(HadCM3) does not reproduce the Atlantic drying.  Likewise the two most recent generations share a
similar latitudinal distribution of rainfall above ~5°N, with a wetter MH over land, peaking at ~2-3
mm day$^{-1}$ at ~11-12°N.  Interestingly, the previous version of the model (HadGEM2-ES) shows the



strongest and most northwardly displaced rainfall peak, as discussed in previous work (e.g. Huag *et al*.
2001, Otto-Bliesner *et al*. 2017, Singarayer *et al*. 2017, Wang *et al*. 2014); the most recent version,
HadGEM3, has lower northward displacement compared to the two older versions of the model. Both
recent versions suggest that the monsoon region extends to ~17°N, above which the differences
between the MH and PI reduce to near zero. In contrast, HadCM3 suggests a generally weaker, but
latitudinally more extensive, monsoon region, suggesting a wetter MH (by ~1 mm day$^{-1}$) as far north
as 20°N and beyond. For the LIG, HadGEM3 is showing a much stronger monsoon region relative to
the *piControl*, compared to HadCM3. However, in terms of extent, similar results are shown to those
for the MH, with HadCM3 showing a generally weaker, but more northwardly displaced, monsoon
region. In this older generation model, positive rainfall anomalies of ~2-3 mm day$^{-1}$ extend as far
north as 17-18°N, whereas in HadGEM3 they fall to ~1 mm day$^{-1}$ at these latitudes.

In terms of the spatial patterns of the West African monsoon, Figure 10 and Figure 11 show the JJA
daily rainfall climatology differences from the same three model generations for the MH and LIG,
respectively. During the MH, consistent with Figure 9, the two most recent simulations generally
agree (RMSE = 0.46 mm day$^{-1}$) and show similar spatial patterns, with a drier equatorial Atlantic
during the MH and then increased rainfall around 10°N (Fig. 10a and b for HadGEM3 and
HadGEM2-ES, respectively). Both simulations also suggest that the increases in rainfall extend
longitudinally across the entire continent, with the largest changes not only occurring across western
and central regions but also further east. In contrast, HadCM3 is less consistent than HadGEM3
(RMSE = 0.53 mm day$^{-1}$) and only suggests a wetter MH over West Africa; moreover, again
consistent with Figure 9, HadCM3 suggests that although the West African monsoon region is
longitudinally narrower, it is latitudinally wider than the other two simulations (Fig. 10c). HadCM3
also differs from the other simulations over the equatorial Atlantic, showing a region of drying that is
not only stronger in magnitude (with the MH being over 5 mm day$^{-1}$ drier than the PI in HadCM3,
compared to ~2-3 mm day$^{-1}$ in the two most recent simulations), but also larger in terms of latitude
and longitude extent (Fig. 10c).

During the LIG, only the most recent and oldest version of the model can be compared, as a LIG
simulation using HadGEM2-ES is unavailable. In Figure 11 there is a noticeable difference between
generations and the level of agreement is the lowest across all simulation combinations (RMSE = 1.57
mm day$^{-1}$), with the most recent HadGEM3 showing greatly increased rainfall across all of northern
Africa, centred on 10°N but extending from ~5°N to almost 20°N and beyond (Fig. 11a), again
consistent with Figure 9. In contrast, and similar to the MH results, in HadCM3 the largest rainfall
increases are confined to Western Africa only, rather than extending longitudinally across the
continent (Fig. 11b). However, in terms of latitudinal extent, HadCM3 is showing weak wet
anomalies all the way to the Mediterranean, whereas the monsoon region diminishes further south (at



~30°N) in HadCM3 and dry anomalies are suggested North of this. Another noticeable difference is
the region of drying, with the most recent generation model placing this over the equatorial Atlantic
(consistent with the MH) but HadCM3 shifting this further east, over most of central Africa (Fig.
11b). The region of equatorial Atlantic drying shown by the more recent versions of the model is
actually wetter during this HadCM3 LIG simulation.

It would therefore appear that, for the MH, whilst there is less difference between the most recent two
configurations of the model (in terms of a more localised West African monsoon region), there
nevertheless has been improvement since the oldest version of the UK's physical climate model. For
the LIG, where unfortunately there is no intermediate generation, it would be reasonable to say that
again considerable change has occurred since the oldest generation model, with the suggestion that,
although HadCM3 is identifying an enhanced monsoon which extends to the Mediterranean (albeit
with very weak anomalies), at lower latitudes it is not showing the level of northward displacement as
the most recent version, apart from in the far western regions.

*3.2.3.2. Rainfall across the Sahara*
Given that the warm climate simulations, and indeed the *piControl*, did not use interactive, but rather
prescribed, vegetation, it is not possible to directly test if the model is reproducing the 'Saharan
greening' that proxy data suggest. For example, Jolly *et al*. (1998a, 1998b) analysed MH pollen
assemblages across northern Africa and suggested that some areas south of 23°N (characterised by
desert today) were grassland and xerophytic woodland/scrubland during the MH (Joussaume *et al*.
1999). To circumvent this caveat, Joussaume *et al*. (1999) developed a method for indirectly
assessing Saharan greening, based on the annual mean rainfall anomaly relative to a given model's
modern simulation. Using the water-balance module from the BIOME3 equilibrium vegetation model
(Haxeltine & Prentice 1996), Joussaume *et al*. (1999) calculated the increase in mean annual rainfall,
zonally averaged over 20°W-30°E, required to support grassland at each latitude from 0 to 30°N,
compared to the modern rainfall at that latitude. This was then used to create maximum and
minimum estimates, within which bounds the model's annual mean rainfall anomaly must lie to
suggest enough of an increase to support grassland (Joussaume *et al*. 1999).

Therefore, an adapted version of Figure 3a in Joussaume *et al*. (1999) is shown here in Figure 12,
which includes the above mean annual rainfall anomalies from not only the current *midHolocene*
simulation, but also all previous MH simulations from CMIP5. Firstly of note is that, despite the
equatorial Atlantic drying that all the models show (seen, for example, in Figure 5), the HadGEM3
*midHolocene* simulation is showing a peak in rainfall further south compared to many other CMIP5
models, suggesting less northward displacement of the rainbelt relative to the other models (Fig. 12).
Concerning the threshold required to support grassland, it is clear that although the current





*midHolocene* simulation is showing an increase in mean annual rainfall further north than some of the
models, including its predecessor HadGEM2-ES, and is just within the required bounds at lower
latitudes (e.g. up to 17°N), north of this the current *midHolocene* simulation is not meeting the
required threshold, neither are any of the other CMIP5 models after ~18°N (Fig. 12).  It would
therefore appear that, although some improvement has been made since CMIP5 and earlier models,
the latest version of the UK's physical climate model it is still unable to reproduce the amount of
rainfall necessary to give the 'Saharan greening' suggested by proxy data during the MH.

**4. SUMMARY AND CONCLUSIONS**
This study has conducted and assessed the mid-Holocene and Last Interglacial simulations using the
latest version of the UK's physical climate model, HadGEM3-GC3.1, comparing the results with
available proxy data, previous versions of the same model, and other models from CMIP's previous
iteration, CMIP5.  Both the *midHolocene* and *lig127k* simulations followed the experimental design
defined in Otto-Bliesner *et al*. (2017) and under the auspices of CMIP6/PMIP4,  Both simulations
were run for a 350-400 year spin-up phase, during which time atmospheric and oceanic equilibrium
was assessed, and once an acceptable level of equilibrium had been reached, the production runs were
started.

Concerning the results from the spin-up phase, comparison to the metrics used to assess the CMIP6
*piControl* suggest that both warm climate simulations reached an acceptable state of equilibrium, in
the atmosphere at least, to allow the production runs to be undertaken.  From these, both simulations
are showing global temperatures consistent with the latitudinal and seasonal distribution of insolation,
and with previous work (e.g. Otto-Bliesner *et al*. 2017).  Globally, whilst both the simulations are
mostly capturing the sign and, in some places, magnitude of change relative to the PI, similar to
previous model simulations this is geographically and seasonally dependent.  It should be noted that
the proxy data (against which the simulations are evaluated) also contain a high level of uncertainty in
both space and time, and so it is encouraging that the simulations are generally reproducing the large-
scale sign of change, if not at an individual location.  Likewise, the behaviour of the West African
monsoon in both simulations is consistent with current understanding (e.g. Huag et al. 2001,
Singarayer et al. 2017, Wang et al. 2014), which suggests a wetter (and possibly latitudinally wider,
and/or northwardly displaced) monsoon during the MH and LIG, relative to the PI.  Regarding model
development in simulating the West African monsoon, although there has been an improvement since
the oldest version of the UK's physical climate model (HadCM3), the two most recent version of the
model yield similar results in terms of both intensity and position.  Lastly, regarding the well-
documented 'Saharan greening' during the MH, results here suggest that the most recent version of
the UK's physical climate model is consistent with all other previous models to date.



In conclusion, the results suggest that the most recent version of the UK's physical climate model is
reproducing climate conditions consistent with the known changes to insolation during these two
warm periods, and is consistent with previous versions of the same model, and other models. Even
though the *lig127k* simulation did not contain any influx of Northern Hemisphere meltwater, shown
by previous work to be a critical forcing in LIG warming, it is still nevertheless showing increased
temperatures in certain regions. A potential caveat of this conclusion, however, is the matter of spin-
up and the fact that neither of the current warm climate simulations were in oceanic equilibrium when
the production runs were undertaken. The production runs were undertaken nevertheless because the
resources required to run for several thousands of years (needed to reach true oceanic equilibrium)
would have been impossible to obtain, but future simulations using this model should endeavour to
obtain a better level of oceanic equilibrium. Another limitation of using this particular version of the
model is that certain processes, such as vegetation and atmospheric chemistry, were prescribed, rather
than allowed to be dynamically evolving. Moreover, for reasons of necessity some of the boundary
conditions were left as PI, such as vegetation, surface like, anthropogenic deforestation and aerosols; a
better simulation might be achieved if these were prescribed for the MH. Processes and boundary
conditions such as these may be of critical importance regarding climate sensitivity during the MH
and the LIG, and therefore ongoing work is underway to repeat both of these experiments using the
most recent version of the UK's Earth Systems model, UKESM1. Here, although the atmospheric
core is HadGEM3, UKESM1 contains many other earth system components (e.g. dynamic
vegetation), and therefore in theory should be able to better reproduce these paleoclimate states.

**DATA AVAILABILITY**
For the MH reconstructions, the data can be found within the Supplementary Online Material of
Bartlein *et al*. (2011), at https://link.springer.com/article/10.1007/s00382-010-0904-1. For the LIG
reconstructions, the data can be found within the Supplementary Online Material of Capron *et al*.
(2017), at https://www.sciencedirect.com/science/article/pii/S0277379117303487?via%3Dihub, and
the Supplementary Online Material of Hoffman *et al*. (2017), at
https://science.sciencemag.org/content/suppl/2017/01/23/355.6322.276.DC1. The model simulations
will be uploaded in early 2020 to the Earth System Grid Federation (ESGF) WCRP Coupled Model
Intercomparison Project (Phase 6), but are not yet available. The simulations are currently available
by directly contacting the lead author.

**COMPETING INTERESTS**
The authors declare that they have no conflict of interest.

**AUTHOR CONTRIBUTION**



CJRW conducted the *midHolocene* simulation, carried out the analysis, produced the figures, wrote the majority of the manuscript, and led the paper. MVG conducted and provided the *lig127k* simulation, and contributed to some of the analysis and writing. EC provided the proxy data, and contributed to some of the writing. IMV provided the HadCM3 LIG simulation. PJV provided the HadCM3 MH simulation. JS contributed to some of the writing. All authors proofread the manuscript and provided comments.

**ACKNOWLEDGEMENTS**

CJRW acknowledges the financial support of the UK Natural Environment Research Council-funded SWEET project (Super-Warm Early Eocene Temperatures), research grant NE/P01903X/1. CJRW also acknowledges the financial support of the Belmont-funded PACMEDY (PAlaeo-Constraints on Monsoon Evolution and Dynamics) project, as does JS. MVG and LCS acknowledge the financial support of the NERC research grants NE/P013279/1 and NE/P009271/1. EC acknowledges financial support from the ChronoClimate project, funded by the Carlsberg Foundation.





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





**LIST OF TABLES**
Table 1 - Astronomical parameters and atmospheric trace gas concentrations used in HadGEM3
*midHolocene* and *lig127k* simulations

Table 2 - Trends (per century) in global mean measures of climate equilibrium for the last hundred
years of the simulations, adapted from and including *piControl* results from Menary *et al*. (2018).
Note - For temperature, Menary *et al*. (2018) provide SAT.  For OceTemp and OceSal, these were
calculated using the full-depth ocean for the *piControl*, whereas in the other two simulations these
fields were calculated down to a depth of 1045m

Table 3 - Global 1.5 m air temperature means and anomalies from HadGEM3 *piControl*,
*midHolocene* and *lig127k* production runs (100-year climatology)

Table 4 - RMSE values for *midHolocene* and *lig127k* production runs (100-year climatology) versus:
a) proxy data from Bartlein *et al*. (2011) for the MH and Capron *et al*. (2017) / Hoffman *et al*. (2017)
for the LIG; b) MH and LIG simulations from previous versions of UK model.  Regarding the proxy
data comparisons in a), for JAS the simulated SST anomalies are compared to Northern Hemisphere
summer reconstructions and for JFM the simulated SST anomalies are compared to Southern
Hemisphere summer reconstructions.  Note that, as shown in Figure 8, proxy locations show SST over
ocean and SAT over Greenland/Antarctica; to calculate RMSE values, however, only SST data were
used

**LIST OF FIGURES**
Figure 1 - Latitude-month insolation (incoming SW radiative flux) anomalies: a) *midHolocene -*
*piControl*; b) *lig127k - piControl*

Figure 2 - Annual global mean atmospheric fields from HadGEM3 *piControl*, *midHolocene* and
*lig127k* spin-up phases: a) 1.5 m air temperature; b) TOA.  Thin lines in b) show annual TOA, thick
lines show 11-year running mean

Figure 3 - Annual global mean oceanic fields from HadGEM3 *piControl*, *midHolocene* and *lig127k*
spin-up phases: a) OceTemp down to 1045m; b) OceSal down to 1045m

Figure 4 – 1.5 m air temperature climatology differences, HadGEM3 *midHolocene* and *lig127k*
production runs versus HadGEM3 *piControl* production run, 100-year climatology from each: a)
*midHolocene – piControl*, JJA; b) *midHolocene – piControl*, DJF; c) *lig127k – piControl*, JJA; d)
*lig127k – piControl*, DJF




Figure 5 – JJA rainfall & 850mb wind climatology differences, HadGEM3 *midHolocene* and *lig127k*
production runs versus HadGEM3 *piControl* production run, 100-year climatology from each: a)
*midHolocene – piControl*, JJA; b) *lig127k – piControl*; c) *lig127k – midHolocene*

Figure 6 – JJA rainfall differences by latitude, averaged over West Africa (20°W-30°E, including both
land and ocean points), HadGEM3 *midHolocene* and *lig127k* production runs versus HadGEM3
*piControl* production run, 100-year climatology from each year

Figure 7 – Simulated versus proxy MAT and MAP anomalies.  Left-hand side panels show simulated
gridded anomalies from HadGEM3 (*midHolocene* production run – *piControl* production run, 100-
year climatology from each), right-hand side panels show proxy data from Bartlein *et al*. (2011) (MH
– preindustrial).  Proxy data locations are projected onto model grid: a) Simulated MAT; b) Proxy
MAT; c) Simulated MAP; d) Proxy MAP

Figure 8 – Simulated versus proxy SST anomalies.  Background gridded data show simulated
anomalies (*lig127k* production run – *piControl* production run) from HadGEM3 (100-year
climatology), circles show proxy data (LIG – preindustrial) from Capron *et al*. (2017) and triangles
show proxy data (LIG – preindustrial) from Hoffman *et al*. (2017).  Proxy data locations are projected
onto model grid: a) Annual data; b) Northern Hemisphere summer (JAS); c) Southern Hemisphere
summer (JFM).  Note that proxy locations show SST over ocean and SAT over Greenland/Antarctica

Figure 9 – JJA daily rainfall climatology differences (MH and LIG-PI) by latitude, averaged over
West Africa (20°W-30°E, including both land and ocean points), for the various generations of the
UK's physical climate model, 100-year climatology from each (50-year climatology for HadCM3
LIG).  Solid lines show MH simulations, dotted lines show LIG simulations.  Note that due to the low
spatial resolution in HadCM3, values in between latitude points have been interpolated

Figure 10 – JJA daily rainfall climatology differences (MH-PI) for the various generations of the
UK's physical climate model, 100-year climatology from each: a) HadGEM3; b) HadGEM2-ES; c)
HadCM3

Figure 11 – JJA daily rainfall climatology differences (LIG-PI) for the various generations of the
UK's physical climate model, 100-year climatology from HadGEM3, 50-year climatology from
HadCM3: a) HadGEM3; b) HadCM3



Figure 12 – Annual mean rainfall over West Africa, zonally averaged from 20°W-30°E, HadGEM3
and CMIP5 *midHolocene* production run minus corresponding *piControl* production runs, 100-year
climatology.  Solid line shows HadGEM3, dotted lines show CMIP5 simulations.  Grey dashes show
maximum and minimum bounds of the increase in rainfall required to support grassland at each
latitude, within which simulations must lie if producing enough rainfall to support grassland



|  | *piControl* | *midHolocene* | *lig127k* |
|---|---|---|---|
| **Astronomical parameters** | | | |
| **Eccentricity** | 0.016764 | 0.018682 | 0.039378 |
| **Obliquity** | 23.459 | 24.105° | 24.04° |
| **Perihelion-180°** | 100.33 | 0.87° | 275.41° |
| **Date of vernal equinox** | March 21 at noon | March 21 at noon | March 21 at noon |
| **Trace gases** | | | |
| **CO₂** | 284.3 ppm | 264.4 ppm | 275 ppm |
| **CH₄** | 808.2 ppb | 597 ppb | 685 ppb |
| **N₂O** | 273 ppb | 262 ppb | 255 ppb |
| **Other GHG gases** | CMIP DECK *piControl* | CMIP DECK *piControl* | CMIP DECK *piControl* |

Table 1 - Astronomical parameters and atmospheric trace gas concentrations used in HadGEM3
simulations

| Variable | *piControl* | *midHolocene* | *lig127k* |
|---|---|---|---|
| **TOA (W m²)** | -0.002 | -0.05 | -0.06 |
| **1.5 m air temp (°C)** | 0.03 | -0.06 | -0.16 |
| **OceTemp (°C)** | 0.035 | 0.0002 | 0.02 |
| **OceSal (psu)** | 0.0001 | -0.007 | 0.006 |

Table 2 - Trends (per century) in global mean measures of climate equilibrium for the last hundred
years of the simulations, adapted from and including *piControl* results from Menary *et al*. (2018).
Note - For temperature, Menary *et al*. (2018) provide SAT.  For OceTemp and OceSal, these were
calculated using the full-depth ocean for the *piControl*, whereas in the other two simulations these
fields were calculated down to a depth of 1045m

| Time period | Means (°C) | | | Anomalies (°C) | |
|---|---|---|---|---|---|
| | *piControl* | *midHolocene* | *lig127k* | *midHolocene – piControl* | *lig127k – piControl* |
| **Annual** | 13.8 | 13.67 | 14.29 | -0.12 | 0.49 |
| **JJA** | 15.68 | 15.75 | 17.37 | 0.07 | 1.69 |
| **DJF** | 11.86 | 11.55 | 11.39 | -0.31 | -0.47 |

Table 3 - Global 1.5 m air temperature means and anomalies from HadGEM3 *piControl*,
*midHolocene* and *lig127k* production runs (100-year climatology)






| Metric | a) Simulations versus proxy data | | | | | |
|---|---|---|---|---|---|---|
| | **MH** | **LIG** | | | | |
| **MAT (°C)** | 2.45 | *Capron et al.* (2017) | | | *Hoffman et al.* (2017) | |
| *No. of proxy locations* | *638* | | | | | |
| **MAP (mm year⁻¹)** | 280 | | | | | |
| *No. of proxy locations* | *651* | | | | | |
| **SST (°C)** | | **Yearly** | **JAS** | **JFM** | **Yearly** | **JAS** | **JFM** |
| | | 2.44 | 3.11 | 1.94 | 2.94 | 2.06 | 4.24 |
| *No. of proxy locations* | | *7* | *24* | *15* | *86* | *12* | *6* |
| | **b) Simulations versus simulations** | | | | | |
| | **MH** | | **LIG** | | | |
| | **HadGEM2-ES v HadGEM3** | **HadCM3 v HadGEM3** | **HadCM3 v HadGEM3** | | | |
| **JJA rainfall (mm day⁻¹)** | 0.46 | 0.53 | 1.57 | | | |


Table 4 - RMSE values for *midHolocene* and *lig127k* production runs (100-year climatology) versus:
a) proxy data from Bartlein *et al.* (2011) for the MH and Capron *et al.* (2017) / Hoffman *et al.* (2017)
for the LIG; b) MH and LIG simulations from previous versions of UK model.  Regarding the proxy
data comparisons in a), for JAS the simulated SST anomalies are compared to Northern Hemisphere
summer reconstructions and for JFM the simulated SST anomalies are compared to Southern
Hemisphere summer reconstructions.  Note that, as shown in Figure 8, proxy locations show SST over
ocean and SAT over Greenland/Antarctica; to calculate RMSE values, however, only SST data were
used






**FIGURES**

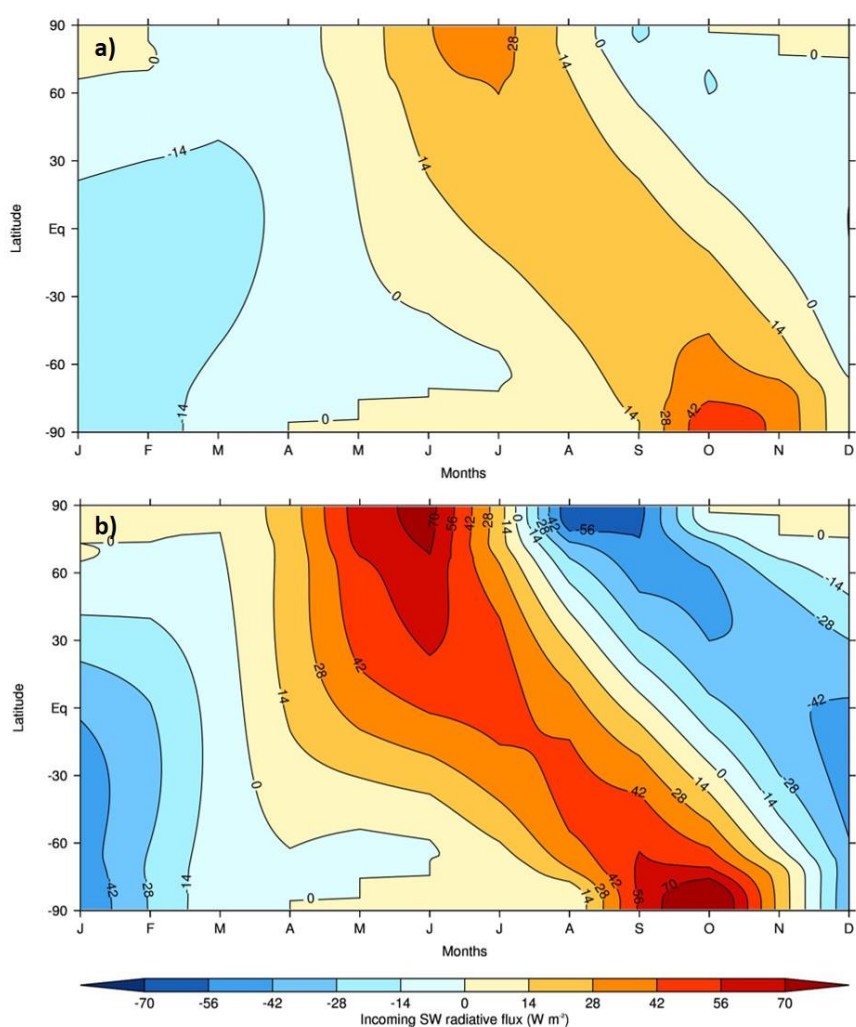


Figure 1 - Latitude-month insolation (incoming SW radiative flux) anomalies: a) *midHolocene -*
*piControl*; b) *lig127k - piControl*






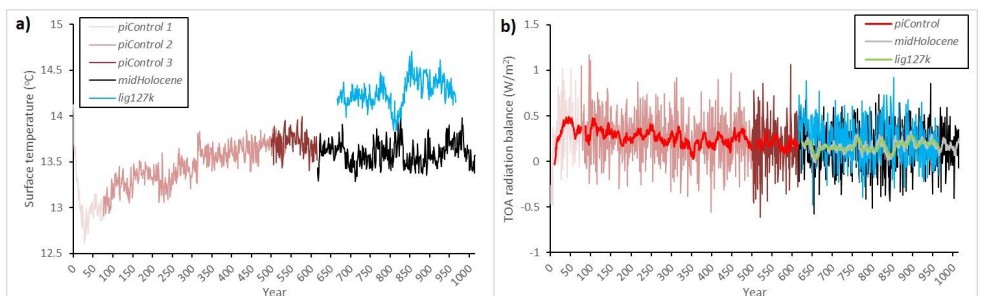


Figure 2 - Annual global mean atmospheric fields from HadGEM3 *piControl*, *midHolocene* and
*lig127k* spin-up phases: a) 1.5 m air temperature; b) TOA.  Thin lines in b) show annual TOA, thick
lines show 11-year running mean





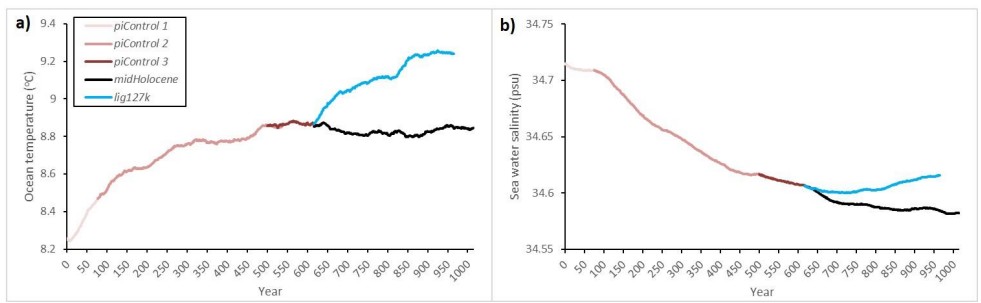

Figure 3 - Annual global mean oceanic fields from HadGEM3 *piControl*, *midHolocene* and *lig127k*

spin-up phases: a) OceTemp down to 1045m; b) OceSal down to 1045m





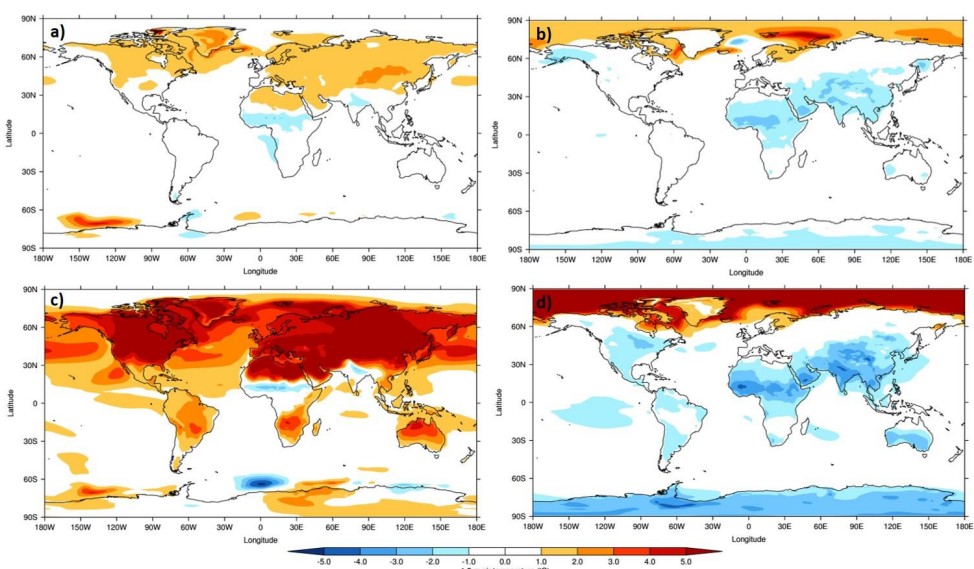


Figure 4 – 1.5 m air temperature climatology differences, HadGEM3 *midHolocene* and *lig127k*
production runs versus HadGEM3 *piControl* production run, 100-year climatology from each: a)
*midHolocene – piControl*, JJA; b) *midHolocene – piControl*, DJF; c) *lig127k – piControl*, JJA; d)
*lig127k – piControl*, DJF



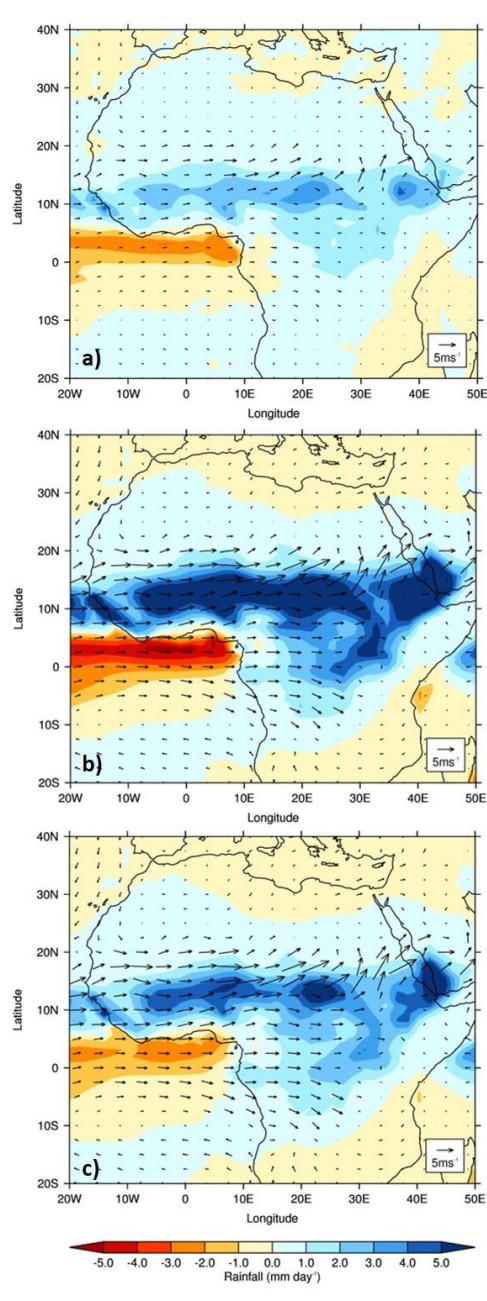


Figure 5 – JJA rainfall & 850mb wind climatology differences, HadGEM3 *midHolocene* and *lig127k*
production runs versus HadGEM3 *piControl* production run, 100-year climatology from each: a)
*midHolocene – piControl*, JJA; b) *lig127k – piControl*; c) *lig127k – midHolocene*



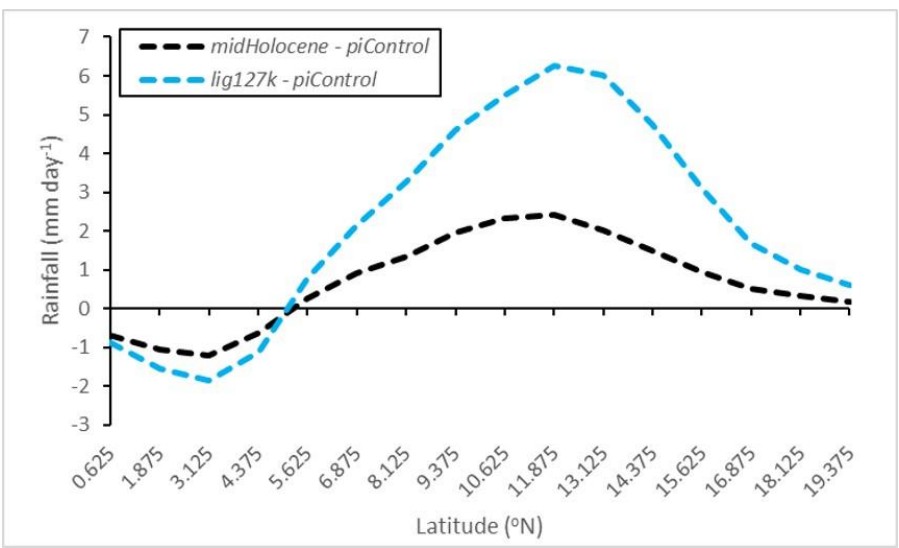


Figure 6 – JJA rainfall differences by latitude, averaged over West Africa (20°W-30°E, including both
land and ocean points), HadGEM3 *midHolocene* and *lig127k* production runs versus HadGEM3
*piControl* production run, 100-year climatology from each year




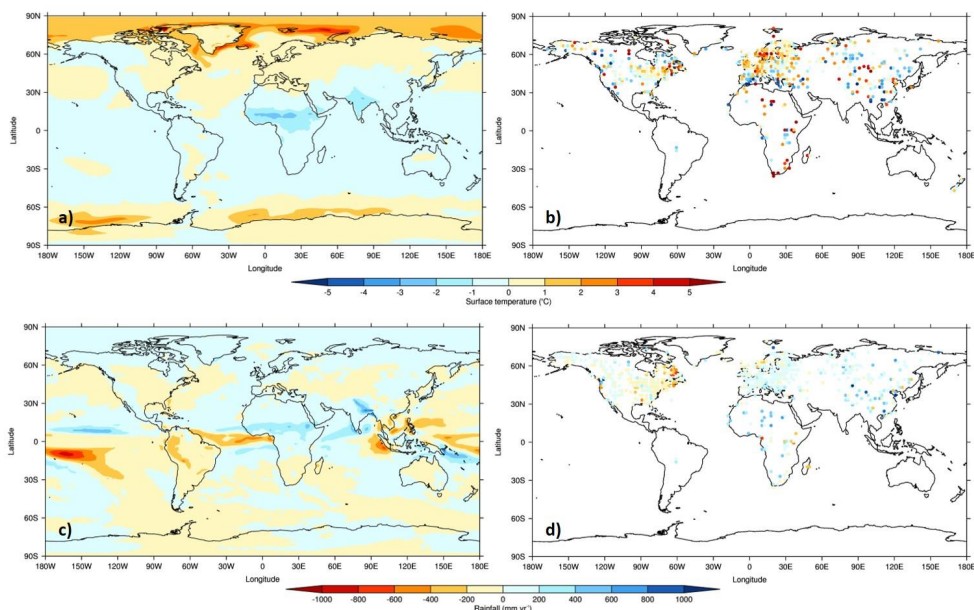


Figure 7 – Simulated versus proxy MAT and MAP anomalies. Left-hand side panels show simulated
gridded anomalies from HadGEM3 (*midHolocene* production run – *piControl* production run, 100-
year climatology from each), right-hand side panels show proxy data from Bartlein *et al*. (2011) (MH
– preindustrial). Proxy data locations are projected onto model grid: a) Simulated MAT; b) Proxy
MAT; c) Simulated MAP; d) Proxy MAP

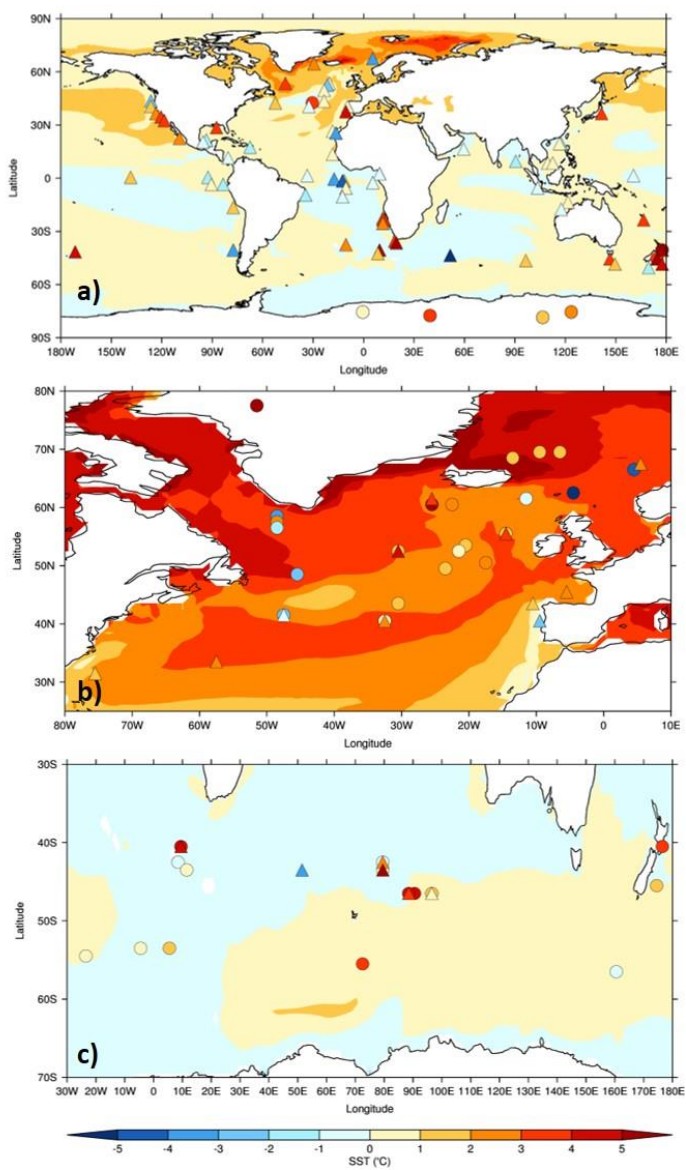


Figure 8 – Simulated versus proxy SST anomalies. Background gridded data show simulated
anomalies (lig127k production run – piControl production run) from HadGEM3 (100-year
climatology), circles show proxy data (LIG – preindustrial) from Capron et al. (2017) and triangles
show proxy data (LIG – preindustrial) from Hoffman et al. (2017). Proxy data locations are projected
onto model grid: a) Annual data; b) Northern Hemisphere summer (JAS); c) Southern Hemisphere
summer (JFM). Note that proxy locations show SST over ocean and SAT over Greenland/Antarctica





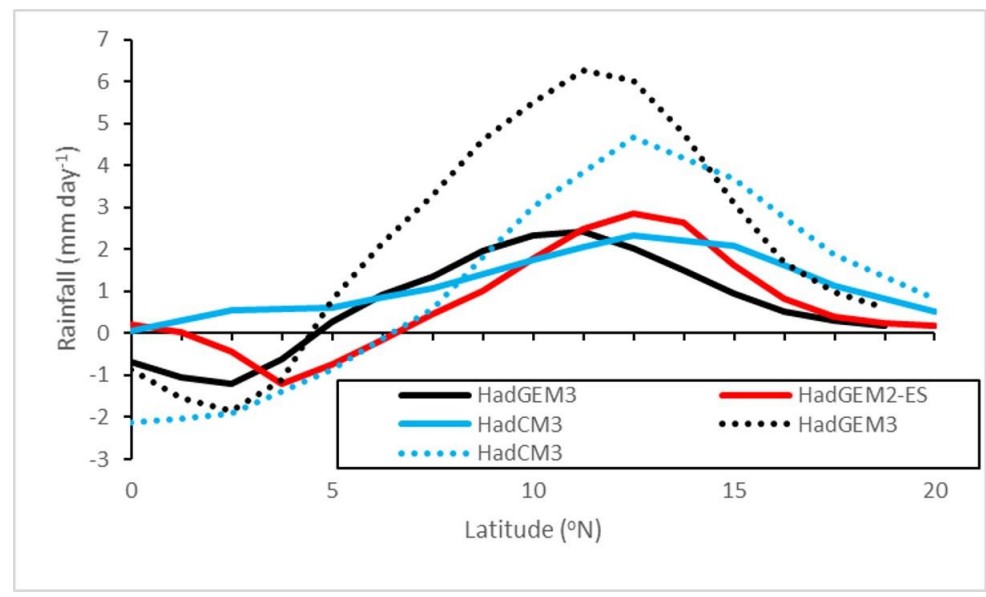

Figure 9 – JJA daily rainfall climatology differences (MH and LIG-PI) by latitude, averaged over
West Africa (20°W-30°E, including both land and ocean points), for the various generations of the
UK's physical climate model, 100-year climatology from each (50-year climatology for HadCM3
LIG). Solid lines show MH simulations, dotted lines show LIG simulations. Note that due to the low
spatial resolution in HadCM3, values in between latitude points have been interpolated

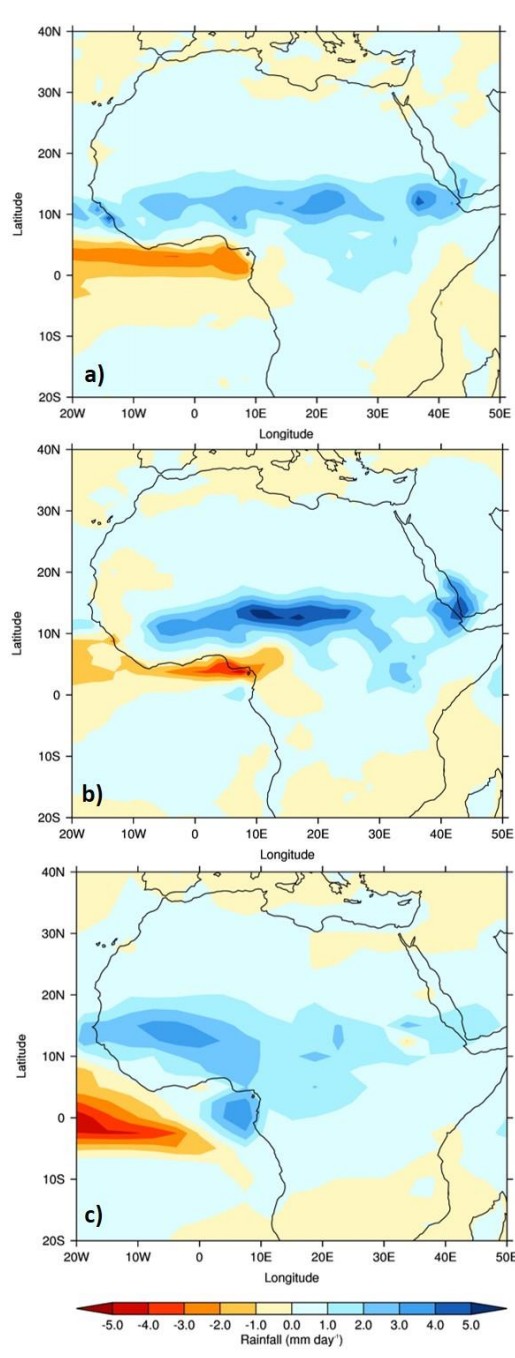


Figure 10 – JJA daily rainfall climatology differences (MH-PI) for the various generations of the
UK's physical climate model, 100-year climatology from each: a) HadGEM3; b) HadGEM2-ES; c)
HadCM3

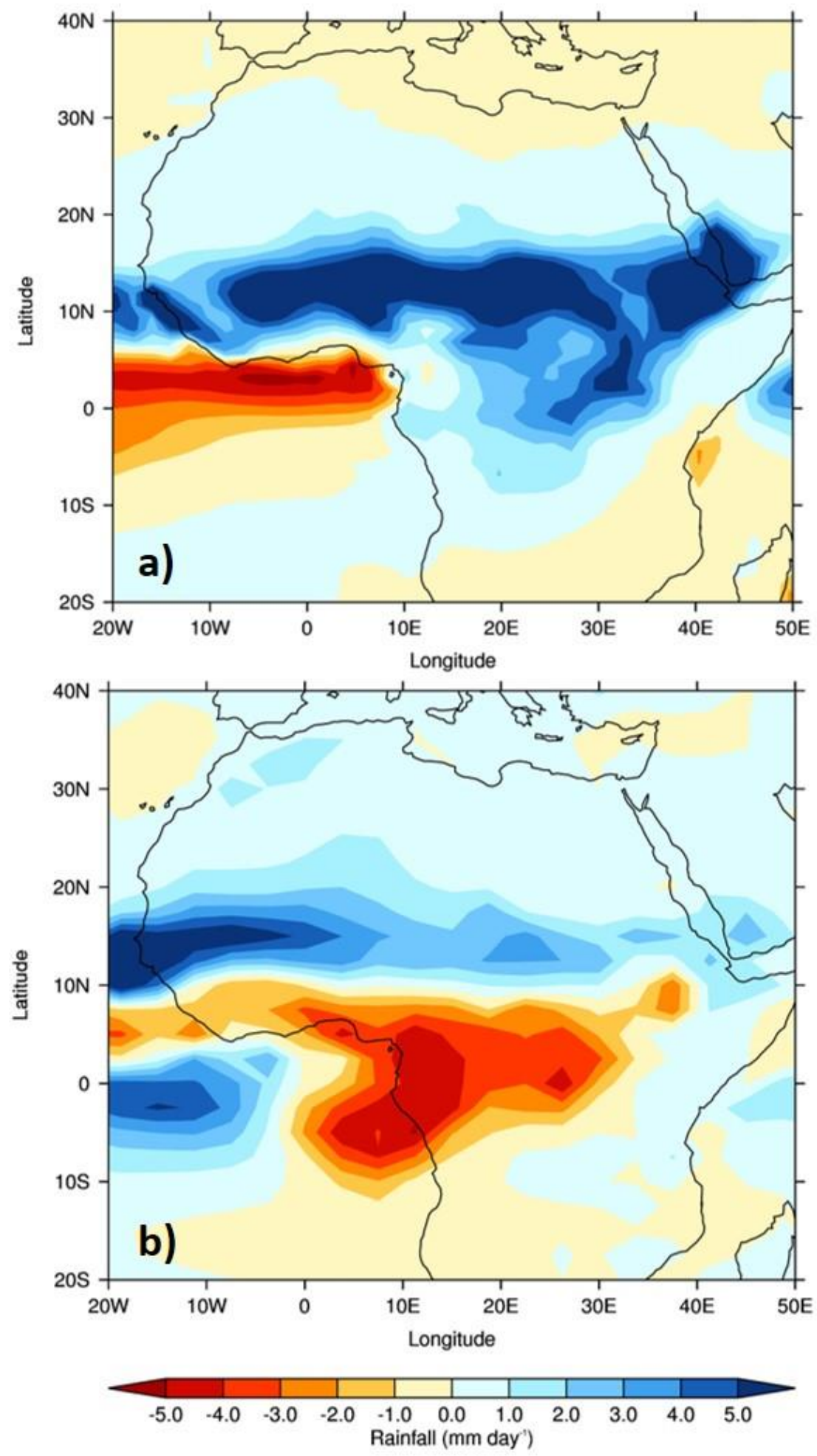

Figure 11 – JJA daily rainfall climatology differences (LIG-PI) for the various generations of the UK's physical climate model, 100-year climatology from HadGEM3, 50-year climatology from HadCM3: a) HadGEM3; b) HadCM3




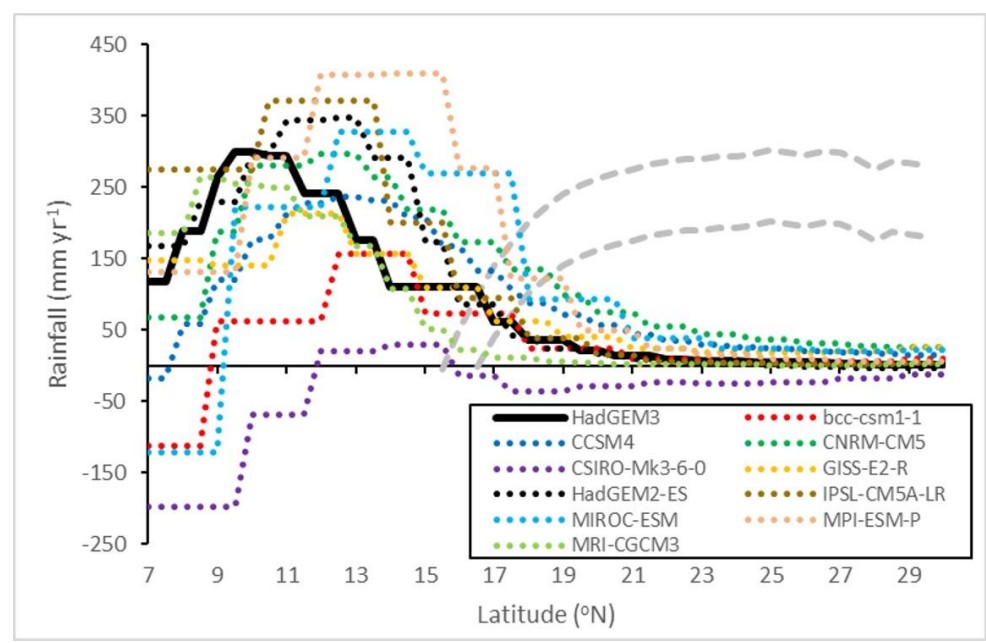


Figure 12 – Annual mean rainfall over West Africa, zonally averaged from 20°W-30°E, HadGEM3
and CMIP5 *midHolocene* production run minus corresponding *piControl* production runs, 100-year
climatology.  Solid line shows HadGEM3, dotted lines show CMIP5 simulations.  Grey dashes show
maximum and minimum bounds of the increase in rainfall required to support grassland at each
latitude, within which simulations must lie if producing enough rainfall to support grassland
