# Peer review of "The UK contribution to CMIP6/PMIP4: mid-Holocene and Last Interglacial experiments with HadGEM3, and comparison to the pre- industrial era and proxy data"

_Climate of the Past, 2019_

## Referee Comment (RC1) · Anonymous Referee #1 · 30 Jan 2020

Review of manuscript by Williams et al. entitled "The UK contribution to CMIP6/PMIP4: mid-Holocene and Last Interglacial experiments with HadGEM3, and comparison to the pre-industrial era and proxy data"

The manuscript presents an extensive description of the new "past warm climate" simulations with the UK climate model and I applaud the effort to compare the results both to proxy-data as well as to previous versions of the same model. However,

in the current form the manuscript is difficult to read as it seems to presents a bit of everything, lacking a clear aim or structure. Moreover, there are several other issues that need to be addressed before I can recommend publication of the manuscript. Below I will detail my main concerns and list all minor and technical comments.

Main concerns:

As mentioned above, it appears to me that the manuscript presents various data sets and experiments that in its current form lack relevance or seem somewhat out of place. This makes that the manuscript lacks a clear common thread, making the manuscript difficult to read. Clarify the relevance of all the presented results and consistently mention all of them in the abstract, introduction, results and conclusion sections. Clarify why you present: a comparison to previous versions of the model; a comparison to CMIP/PMIP results; a comparison with proxy-data. A change to the overall structure of the manuscript as detailed in the following could also improve the flow of the manuscript.

The analysis of temperature on the one hand and precipitation on the other hand are very different. Temperatures are looked at globally, while the analysis of precipitation is solely on Africa. Clarify to the reader why this choice was made. The structure of the manuscript would potentially be improved if the analysis is firstly on global features (mostly temperature, but perhaps also precipitation considering the newly developed Last Interglacial precipitation reconstruction), and secondly zooms in on African precipitation changes.

An extensive description of the spin-up results is given in the manuscript. While in general I appreciate it that such potentially important modelling details are given, the relevance to the rest of the manuscript is not clear to me. Wouldn't it be sufficient

to provide the numbers of the spin-up results in a table and refer to that table in the method section? Is a whole results section for the spin-up results needed, given also that they are not referred to anymore in the remainder of the analysis?

The model-data comparison is limited to temperature proxies while recently a new compilation of precipitation reconstructions for the Last Interglacial has become available (Scussolini et al., 2019) which should be used here as well.

I find the whole analysis of African precipitation changes in the different simulations incomplete and too simplistic, leading to figures and results that are misleading. To improve this situation I suggest the following:
-> Remove the ocean grid cells from the domain over which the analysis is performed. Presenting zonal-mean figures is not appropriate if the results are clearly not zonally homogeneous as is the case around the equator in this analysis.
-> Why is the focus on JJA precipitation over Africa? The results in figure 12 are again annual mean. Be consistent and clarify your reasoning.
-> The authors seem to use the words 'monsoon' and 'ITCZ' interchangeably. While indeed they cannot easily be separated based solely on the analysis of precipitation, they are driven by fundamentally different processes and an attempt should be made to separate the two. An interesting read on this topic is by Nicholson (2009).
-> The results for African precipitation are rather different for the different periods that are considered and for the various versions of the climate model. Provide some analysis as to what drives these differences.

In many figures no measure of the robustness of the results is given. Provide measures to determine if your results are significant or if we are looking at internal variability of the climate system. A possibility would be to use long-term variability within the different simulations (PI, 6k, 127k) to deduce whether the depicted anomalies are outside of the range of this variability.

Minor comments:

Line 48: are surface (0 meter) temperatures discussed in the manuscript? Or do the authors mean SSTs here?

Line 82: these periods are not 'warm' everywhere and all the time. Please clarify.

Line 83: 129-116 is not the period that is discussed in this manuscript. Why not simply 127ka?

Lines 108-112: Clarify that this warming is mainly located at high latitudes.

Line 126: convection in the atmosphere or ocean or both?

Line 127: are the ocean and sea ice models completely new or have parts been updated?

Line 140-141: This division in two subsections (3.1 and 3.2) suggest to me that the two topics are of similar importance while in reality this is certainly not the case, with the results on the spin-up phase being only a small side topic. Consider changing this structure to better represent the importance of the different topics.

[Figure]

Line 141: For me the term productions runs is a little strange, perhaps it is CMIP kind of language, but in the context of a manuscript is doesn't mean much to me.

Line 214: Isn't precipitation impacted by ENSO?

Lines 248-252: I don't think such details (number of output variables) are relevant for a manuscript.

Line 273: what is your definition of 'summer'?

Line 276: You constructed this 127 ka time-slice of the Hoffman et al data? Do you provide this data for future work?

Line 306: Is a trend of 0.16 degrees per century small? Sounds significant to me. Please clarify.

Line 335: "the current two warm climate", what does that mean or refer to?

Line 335: Which newly-available proxy data are you referring to? Did you gather new data? Or do you mean the 127 ka time slice based on the Hoffman et al data?

Line 342: HadGEM3 warm climate simulations?

Line 364: 30 degrees east doesn't sound like west African to me. Please clarify why this domain was chosen, also in light of my main concern on this topic.

[Figure]

Line 369: The wind patterns to me show an increase at nearly all latitudes, is that typical for an ITCZ shift?

Lines 374-377: Do we see the same kind of pattern to the south of the equator, so the South African region?

Line 407: Do proxies suggest a global annual mean warming during the MH?

Line 439: "within the average uncertainty range'? Please clarify this statement.

Line 466: The model is seasonally dependent? What does that mean? Do you mean the comparison of models and data?

Lines 488-492: Why would you compare your results to results from previous model version to see if you get sufficient precipitation over the Sahara to promote vegetation growth?

Lines 515-529: These kind of detailed (small) differences make me wonder whether we are really discussing forced differences or if we are discussing internal variability of the system. Please show statistics to argue either way.

Lines 546-553: When you are talking about an 'improvement' this suggest that we know what 'good' means. What kind of data or proxies do you use to determine 'good' and what is the uncertainty of these estimates?

Line 555: Aren't the paragraphs before already discussing "rainfall across the Sahara"?

Lines 570-583: What is the relationship between vegetation in the Sahara (the topic of this paragraph) and the state of the equatorial Atlantic ('drying'?)? Please clarify.

Lines 570-583: Not only is a vegetation model missing to directly determine whether or not vegetation would grow with the simulated amount of precipitation, but also all vegetation related feedbacks on the climate are missing. Discuss the possible impact of these missing feedbacks on you results.

Lines 616-619: meltwater does not only yield a warming, it usually results in a spatially varying pattern with regions of warming and regions of cooling. Please clarify.

Lines 619-621: Is the length of the spin-up really a potentially important caveat? Do you have evidence to support this?

Line 628: Only MH or both MH and 127ka?

Table 2: If some values are for the full ocean depth and others for the top 1054 meter, can we still compare them? Isn't it comparing apples and oranges?

Table 4: I appreciate the attempt to provide a lot of information, but I find this table very confusing. Perhaps it can be split or rearranged?

Figure 1: Have calendar effects been taken into account when making this figure? Please apply corrections, following for instance the methodology outlined by Bartlein et al. (2019).

Figure 2: There seems to be a gap between the control data and the start of the 127k simulation, is this a real data gap or an error in the figure?

Figure 2: Are the temperatures in the left-hand figure surface or 1.5 meter temperatures?

Figure 2: Consider: 'b) TOA radiation balance'

Figure 2: This figure gives a good idea of the amount of internal variability in the system, which seems considerable in both the MH and lig127k simulations. Use this information to define which of your results are robust with respect to this internal variability. Is it true that variability is larger in the 'warm climates' than it is in de control?

Figure 3: For the control simulation the full depth is used instead of the top 1054 meters according to the main text, please clarify.

Figure 4: These figures show some well-know climate change features, including polar amplification. The mechanisms of such spatial temperature anomaly patterns are not discussed. Provide a discussion or refer to previous work on the topic.

Figure 6: Rainfall anomalies on y-axis?

Figure 6: x-axis values are not easy to read in this format.

Figure 6: consider showing absolute precipitation values because I think those give a much better idea of the width of the wet and dry regions as discussed in the main text.

Figure 6: Can't this figure be combined with figure 9?

Figure 8: Remove the ice core data points if the corresponding modeled surface temperature anomalies are not shown.

Figure 9: What does this figure add that is not already depicted in figures 10 and 11? Can't it be removed?

Figure 12: why are the grey dashes that show required rainfall for grassland growth only start from 16 degrees north?

Figure 12: Rainfall anomalies on y-axis?

Figure 12: Why are anomalies shown? Doesn't the threshold to support grassland depend on the absolute amount of precipitation?

Technical comments:

Line 41: are similar

Line 41: period, but

Line 53: generations of the same

Line 121: therefore in the

Line 146: consider removing "indeed"

Line 149: double space before "Full"?

Line 186: tuning of

Line 201: including a reduction of the temperature bias in many regions

Line 221: remove comma after 'design'

Line 244: Too many brackets

Line 272: annual mean surface

Line 298: radiation balance?

Line 384: in the core monsoon region?

Line 395: 'recent', what do you mean?

Line 399: what kind of uncertainty in simulated anomalies are you referring to, please clarify.

Line 400: remove 'often'

Line 437: small number of reconstructions?

Line 449: remove double comma

Line 499: refer to figure 10?

Line 505: smaller northward displacement?

Line 590: 'auspices', not sure if that is the right word for it.

Line 590: replace comma by a dot.

Line 591: remove 'time'?

Line 592: were assessed?

Line 603: 'time', are we talking seasons or different geological intervals?

Line 626: 'necessity' is perhaps a bit too strong in this context.

Line 1007: better not to use the '' symbol.

Line 1008: for each

Line 1018: in this caption and some others the words 'simulated gridded anomalies' are used. This sounds a little double to me since nearly all climate models work on spatial grids so the output is per definition also gridded.

Lines 1024-1027: Is there no overlap between these two data sets? No single core was used in both of them?

Line 1033: erroneous bracket?

References:

Bartlein, P. J., and S. L. Shafer (2019), Paleo calendar-effect adjustments in time-slice

and transient climate-model simulations (PaleoCalAdjust v1.0): impact and strategies for data analysis, Geosci. Model Dev., 12(9), 3889-3913.

Sharon E. Nicholson (2009). A revised picture of the structure of the "monsoon" and land ITCZ over West Africa. Climate Dynamics volume 32.

---

## Referee Comment (RC2) · Anonymous Referee #2 · 14 Feb 2020

Comments on the manuscript entitled Williams, C. J. R., Guarino, M.-V., Capron, E., Malmierca-Vallet, I., Singarayer, J. S., Sime, L. C., Lunt, D. J., and Valdes, P. J.: The UK contribution to CMIP6/PMIP4: mid-Holocene and Last Interglacial experiments with HadGEM3, and comparison to the pre-industrial era and proxy data, Clim. Past Discuss., https://doi.org/10.5194/cp-2019-160, in review, 2020.

The paper describes results from two simulations using the latest version of the UK's physical climate model, HadGEM3-GC3.1; the mid-Holocene ($\sim$6 ka) and Last Interglacial ($\sim$127 ka) simulations, both conducted under the auspices of CMIP6/PMIP4.

[Figure]

Based on three model experiments, this paper presents the response of several climatic variables, especially temperature and precipitation, to changes in insolation and greenhouse gases. Inter-model and model-proxy comparison are also included, but they are not in-depth. It is worth to do this effort as many modelling studies have already been made on the mid-Holocene and Last Interglacial. It is critical to present new findings and methods to make the paper more attractive. At the current stage, there are several major weaknesses from which the paper suffers:

1. It shall be elaborated what is new in this paper in terms of method, result and conclusion as compared to previous studies. Data model comparison in SST data and the question of seasonality could be more elaborated. It is not understandable that the SST comparison has not been performed on the MH experiment, although the data quality is higher and especially the dating uncertainty is much lower. Uncertainty is mentioned quite often, but not really elaborated. For the LIG, one could follow ideas outlined in Pfeiffer and Lohmann (CP) dealing with seasons and dating. For the MH, several data sets are available (e.g. Alkenone and Mg/Ca), again with uncertainties in the season or recorder depth.

2. The paper is too descriptive and focuses only on simulated temperature and precipitation. As a special contribution to CMIP6/PMIP4 is based on a single model, I would expect more comprehensive analysis, like the atmospheric and oceanic circulation, ocean states, and the potential relationship or mechanisms between different components. With such I believe the paper will meet the high standard of CP.

3. The authors show precipitation only for Africa. As a paper contributing to the CMIP6/PMIP4, it shall show the model behavior on global rather than regional scale.

More specific comments:

1. Lines 94-104: This paragraph describes the previous studies on the modeled and observed MH and LIG states, which I find is too brief. As there are so many modelling studies and proxy papers, and this is directly linked to the present manuscript, thus I

suggest to make more complete references. It is suggested to split the texts into two paragraphs, one describing the previous simulation results, the other the proxy issues.

2. Lines 106-108: The authors mention that the past warming are indeed different from future warming, as they are driven by quite different thermal forcing mechanisms, orbital parameters and greenhouse gases. I suggest to also mention that, i) the orbital forcing is shortwave and greenhouse gases are related to mainly the longwave radiation flux, ii) difference in orbital parameters leads to uneven horizontal and seasonal changes, but greenhouse gases can cause more uniform anomalies. Furthermore: iii) It is helpful to know the changes of greenhouse gases between MH/LIG and PI are equal to how much radiation flux anomalies? How to calculate such anomalies based on CO2 changes can be found in some papers (e.g., Myhre, et al. 1998, GRL).

3. Lines 161-203 Too detailed information in terms of the changes in model version is give here. I would recommend to simplify the text and to show what aspect/process can be improved in the newest model version. Details could be provided as supplementary material.

4. Lines 205-209: The sensitivity and control experiments are performed on different platforms. I worry about how different the simulated climate can be. If possible, one shall show in the supplement the anomalies of surface temperature based on the same experiment

5. Table 2 and Fig. 2a, the 1.5 m air temperature of LIG still show significant trend in the final years. Could you please show a trend map to check where such trend mainly occurs? Does it happen in the region of interest?

6. Lines 297-323: I think it is not so necessary to describe the spin-up in such a detail. Just show the tables, and I also recommend to put Fig. 2 and Fig. 3 into the supplement.

7. Fig. 4 and Fig. 5: Perform a Student's t-test to identify in which regions the anomalies are significant and which regions related to internal variabilities. Given the relatively short length of the MH and LIG experiments, it is very important to do so.

8. Line 334 'and'=>', and'

9. Lines 336-337: 'in order to' => 'to'

10. Line 342: Title is confusing. The CMIP6 HadGEM3 simulations include the PI, right?

11. Line 351: 'central' => 'Central'

12. Line 359 and a lot of other places in the paper: please make the experiment name consistent throughout the paper, for example, use either MH or midHolocene, the same for LIG and lig127k, piControl and PI.

13. Line 371: greater land-sea contrast... Is it also the same case in your model? I would recommend to check the moist static energy instead of surface temperature, to also include the aspect of moisture.

14. Lines 374-377: the small anomalies... Again please use Student's t-test. Results discussed in the texts should have a significance level above 95%.

15. Lines 373-374: Comparing Fig. 5a and 5b, I observe no obvious shift in ITCZ, only stronger monsoon rainfall in LIG compared to MH.

16. caption of Fig. 6, 9, and 12: Generally West Africa should be within 20W-15E. Why take 20W-30E?

17. Lines 398-400: Please explain where the large uncertainty in proxy comes from.

18. Lines 422-424: Can this underestimation of the warming be used to explain the "Holocene temperature conundrum"? Or, might the "Holocene temperature conundrum" be caused by the fact that most of the proxy locate in regions with positive temperature anomalies? The proxy data represent seasonal or annual mean value? It

might be helpful to discuss these issues. See, e.g. Lohmann et al. (2013, CP) for a comprehensive comparison for SST changes during the Holocene.

19. Line 396: It would be better to clarify here the threshold of RMSE (is there any?) for a reasonable simulation result, in terms of surface temperature, precipitation and sst.

20. Line 447: if => but

21. Fig. 10 and 11: Again, please show significance (t-test).

22. Line 557. The model used prescribed vegetation, and does not consider dust. Please discuss the influence of the lack of interactive vegetation and dust on the Africa monsoon rainfall.

23. Optional: I encourage the author to make a separate discussion section.

References:

Myhre, Gunnar, et al. "New estimates of radiative forcing due to well mixed greenhouse gases." Geophysical research letters 25.14 (1998): 2715-2718.

Lohmann, Gerrit, et al. "A model-data comparison of the Holocene global sea surface temperature evolution." Climate of the Past 9 (2013): 1807-1839.

Pfeiffer, Madlene, and Gerrit Lohmann. "Greenland Ice Sheet influence on Last Inter-glacial climate: global sensitivity studies performed with an atmosphere–ocean general circulation model." Climate of the Past 12 (2016): 1313-1338.

---

## Author Comment (AC1) · 28 May 2020

To whom it may concern,

I extend my sincere appreciation to the Reviewer 1 for their thorough examination of my manuscript, and their detailed and highly constructive comments. I propose to address all of their concerns, both minor and major, so please see attached for a revised manuscript, still with the Track Changes included, to show my proposed modifications. I also attach a tidy version of this manuscript. Please note that the line numbers shown

here relate to the Track Changes version.

Here, I address the reviewer's suggestions, comment-by-comment. I have also attached this as a PDF, where the reviewer's comments are italicised and in a smaller font, and my corresponding response follows in a standard font.

I very much hope that my responses will satisfy the reviewer.

Yours faithfully,

Dr Charles JR Williams, and co-authors

—   REVIEWER 1

MAIN CONCERNS:

As mentioned above, it appears to me that the manuscript presents various data sets and experiments that in its current form lack relevance or seem somewhat out of place. This makes that the manuscript lacks a clear common thread, making the manuscript difficult to read. Clarify the relevance of all the presented results and consistently mention all of them in the abstract, introduction, results and conclusion sections. Clarify why you present: a comparison to previous versions of the model; a comparison to CMIP/PMIP results; a comparison with proxy-data. A change to the overall structure of the manuscript as detailed in the following could also improve the flow of the manuscript.

The manuscript has now been restructured. In short, the section on the spin-up results has been reduced and moved into the Methodology section (with the figures included in the Supplementary Material), and the Results section has been restructured to focus primarily on two measures: firstly a simulation comparison to assess changes relative to the same model's PI simulation, and secondly a model-data comparison in which the most recent version of the simulations are compared to both previous generations of the same model and newly-available proxy data. Importantly, all versions of the same model are now compared against all available proxy data. The section focusing on the

Saharan greening has been reduced and is now discussed only briefly at the end, with the figure being moved into the Supplementary Material. Following the reviewer's comment above, this new structure is consistently mentioned in the abstract, introduction, results and conclusion sections.

The analysis of temperature on the one hand and precipitation on the other hand are very different. Temperatures are looked at globally, while the analysis of precipitation is solely on Africa. Clarify to the reader why this choice was made. The structure of the manuscript would potentially be improved if the analysis is firstly on global features (mostly temperature, but perhaps also precipitation considering the newly developed Last Interglacial precipitation reconstruction), and secondly zooms in on African precipitation changes.

This structure has now been modified, with an additional figure showing global precipitation changes (Figure 3) as well as appropriate discussion (lines 521-534) so that, when discussing the most recent simulations, both temperature and precipitation are considered at the global scale, before zooming into Africa as an example of monsoon changes.

An extensive description of the spin-up results is given in the manuscript. While in general I appreciate it that such potentially important modelling details are given, the relevance to the rest of the manuscript is not clear to me. Wouldn't it be sufficient to provide the numbers of the spin-up results in a table and refer to that table in the method section? Is a whole results section for the spin-up results needed, given also that they are not referred to anymore in the remainder of the analysis?

This section has now been significantly reduced, and moved to a subsection of the Methodology section (lines 394-429). The results are summarised in Table 2 and referred to here, and the figures (showing examples of timeseries for atmospheric and oceanic equilibrium) have been moved into the Supplementary Material (SM2 and SM4).

The model-data comparison is limited to temperature proxies while recently a new compilation of precipitation reconstructions for the Last Interglacial has become available (Scussolini et al., 2019) which should be used here as well.

This compilation has now been included in the manuscript, with a new figure (Figure 10) as well as new text (lines 769-790), which we feel further adds to the novelty of the manuscript.

I find the whole analysis of African precipitation changes in the different simulations incomplete and too simplistic, leading to figures and results that are misleading. To improve this situation I suggest the following: -> Remove the ocean grid cells from the domain over which the analysis is performed. Presenting zonal-mean figures is not appropriate if the results are clearly not zonally homogeneous as is the case around the equator in this analysis. -> Why is the focus on JJA precipitation over Africa? The results in figure 12 are again annual mean. Be consistent and clarify your reasoning. -> The authors seem to use the words 'monsoon' and 'ITCZ' interchangeably. While indeed they cannot easily be separated based solely on the analysis of precipitation, they are driven by fundamentally different processes and an attempt should be made to separate the two. An interesting read on this topic is by Nicholson (2009). -> The results for African precipitation are rather different for the different periods that are considered and for the various versions of the climate model. Provide some analysis as to what drives these differences.

All of the above points have now been addressed: -> The zonal mean figure has now had all ocean grid cells removed -> An explanation is given for why JJA precipitation is considered over Africa (i.e. because it is the primary wet season) versus annual precipitation in figure 12 (which has now been moved to the Supplementary Material) i.e. to be consistent with the proxy data-based threshold (lines 546-551) and (lines 926-927) -> The confusion between the words "monsoon" and "ITCZ" has been clarified, by sticking to terms such as "rainfall maxima" or "rainbelt" (lines 555-565) -> The reason for the difference in African precipitation between the different periods has been made

more clear (lines 488-603), and a possible reason for one of the differences between versions of the model has been added (lines 817-820).

In many figures no measure of the robustness of the results is given. Provide measures to determine if your results are significant or if we are looking at internal variability of the climate system. A possibility would be to use long-term variability within the different simulations (PI, 6k, 127k) to deduce whether the depicted anomalies are outside of the range of this variability.

This has now been addressed, with the global temperature and precipitation figures of the most recent simulations (Figure 2 and Figure 3) now showing the results of a Students t-test (at the 99% confidence level). The same measure of statistical significance was not included when comparing the most recent simulations with previous versions of the same model and proxy data because stippling would make the proxy data locations harder to visualise.

MINOR COMMENTS:

Line 48: are surface (0 meter) temperatures discussed in the manuscript? Or do the authors mean SSTs here?

This line has now been addressed, to differentiate between 1.5m air temperature and SST, both of which are used in different places (lines 49-69)

Line 82: these periods are not 'warm' everywhere and all the time. Please clarify.

This has now been clarified, emphasising that they are two interglacial simulations representing the most recent warm periods (particularly in the Northern Hemisphere) (lines 106)

Line 83: 129-116 is not the period that is discussed in this manuscript. Why not simply 127ka?

This has now been clarified (lines 107)

Lines 108-112: Clarify that this warming is mainly located at high latitudes.

This has now been corrected (lines 169)

Line 126: convection in the atmosphere or ocean or both?

This has now been clarified, to confirm that it should be atmospheric convection (lines 190)

Line 127: are the ocean and sea ice models completely new or have parts been updated?

This has now been corrected, to confirm that these models are not completely new, but rather updated (lines 191)

Line 140-141: This division in two subsections (3.1 and 3.2) suggest to me that the two topics are of similar importance while in reality this is certainly not the case, with the results on the spin-up phase being only a small side topic. Consider changing this structure to better represent the importance of the different topics.

This is now been corrected as per the comment above, with significantly less emphasis being given to the spin up section (which is now in the Methodology section and summarised in Table 2, when the figures in the Supplementary Material (see above comment) (lines 394-429)

Line 141: For me the term productions runs is a little strange, perhaps it is CMIP kind of language, but in the context of a manuscript is doesn't mean much to me.

This has been removed at this line, and has been clarified when used elsewhere by a new Terminology subsection (section 2.1.1) (lines 216-226)

Line 214: Isn't precipitation impacted by ENSO?

This reference to ENSO has been removed

Lines 248-252: I don't think such details (number of output variables) are relevant for a

manuscript.

These have been removed

Line 273: what is your definition of 'summer'?

This has been clarified, to confirm 'summer' here = July September (lines 362)

Line 276: You constructed this 127 ka time-slice of the Hoffman et al data? Do you provide this data for future work?

Yes, that is correct. A sentence has been added into the Data availability section, detailing access (lines 1014-1016)

Line 306: Is a trend of 0.16 degrees per century small? Sounds significant to me. Please clarify.

The reviewer is correct, so the ambiguous word "small" has been removed

Line 335: "the current two warm climate", what does that mean or refer to?

This term has now been clarified in the new Terminology subsection (section 2.1.1), see above comment

Line 335: Which newly-available proxy data are you referring to? Did you gather new data? Or do you mean the 127 ka time slice based on the Hoffman et al data?

This has now been clarified (lines 477)

Line 342: HadGEM3 warm climate simulations?

This has now been clarified (lines 488)

Line 364: 30 degrees east doesn't sound like west African to me. Please clarify why this domain was chosen, also in light of my main concern on this topic.

The domain used to calculate this zonal mean has now been changed, to become 20°W-15°E, with appropriate clarification text (lines 573)

Line 369: The wind patterns to me show an increase at nearly all latitudes, is that typical for an ITCZ shift?

Reference to the ITCZ has now been removed, instead referring to the regions of rainfall maxima, and an appropriate sentence has been added to clarify that the enhanced wind patterns do indeed occur at all latitudes, but especially over regional rainfall maxima (lines 558)

Lines 374-377: Do we see the same kind of pattern to the south of the equator, so the South African region?

This sentence has now been removed, so is no longer problematic

Line 407: Do proxies suggest a global annual mean warming during the MH?

This has now been addressed, to clarify that we don't see a global annual mean warming from proxies, but rather do see warming in many locations (lines 628)

Line 439: "within the average uncertainty range'? Please clarify this statement.

As part of other changes, this sentence has now been removed, as it was ambiguous

Line 466: The model is seasonally dependent? What does that mean? Do you mean the comparison of models and data?

This sentence has now been clarified, to confirm that it should read "the accuracy of the model is seasonally dependent" (lines 799)

Lines 488-492: Why would you compare your results to results from previous model version to see if you get sufficient precipitation over the Sahara to promote vegetation growth?

This entire paragraph has now been removed as part of other changes, so is no longer problematic

Lines 515-529: These kind of detailed (small) differences make me wonder whether

we are really discussing forced differences or if we are discussing internal variability of the system. Please show statistics to argue either way.

This entire paragraph has now been removed as part of other changes, so is no longer problematic. However, as detailed above, significance testing has now been carried out on the most recent warm climate simulations

Lines 546-553: When you are talking about an 'improvement' this suggest that we know what 'good' means. What kind of data or proxies do you use to determine 'good' and what is the uncertainty of these estimates?

This entire paragraph has now been removed as part of other changes, so is no longer problematic. However, as detailed above, all model versions have now be compared against all available proxy data, allowing a quantitative and qualitative determination of 'good'

Line 555: Aren't the paragraphs before already discussing "rainfall across the Sahara"?

Yes, they are. This is therefore now been changed to a more appropriate title (lines 909)

Lines 570-583: What is the relationship between vegetation in the Sahara (the topic of this paragraph) and the state of the equatorial Atlantic ('drying'?)? Please clarify.

As part of other changes, this paragraph has been rewritten and shortened (and indeed the figure has been moved into the Supplementary Material), and therefore this sentence no longer exists

Lines 570-583: Not only is a vegetation model missing to directly determine whether or not vegetation would grow with the simulated amount of precipitation, but also all vegetation related feedbacks on the climate are missing. Discuss the possible impact of these missing feedbacks on you results.

As part of other changes, this paragraph has been rewritten and shortened. However,

an additional sentence has been added at the end of this paragraph, briefly discussing the current lack of vegetation-related feedbacks (lines 933-935)

Lines 616-619: meltwater does not only yield a warming, it usually results in a spatially varying pattern with regions of warming and regions of cooling. Please clarify.

This has been clarified, to reflect the accurate comments of the reviewer (lines 986-987)

Lines 619-621: Is the length of the spin-up really a potentially important caveat? Do you have evidence to support this?

As part of other changes, this sentence has now been removed

Line 628: Only MH or both MH and 127ka?

This has been clarified (lines 997)

Table 2: If some values are for the full ocean depth and others for the top 1054 meter, can we still compare them? Isn't it comparing apples and oranges?

This has been corrected, such that all the values are for the full ocean depth.

Table 4: I appreciate the attempt to provide a lot of information, but I find this table very confusing. Perhaps it can be split or rearranged?

This has now been corrected, with the table being split into Table 4a and b

Figure 1: Have calendar effects been taken into account when making this figure? Please apply corrections, following for instance the methodology outlined by Bartlein et al. (2019).

Calendar adjustments, both this figure and all subsequent figures involving monthly or seasonal data, have now been applied, following the methodology of Pollard & Reusch (2002) and Marzocchi et al. (2015). This is briefly discussed in the introduction (lines 118-121), with examples of the data on the modern calendar (for comparative purposes) included in the Supplementary Material (SM1).

Figure 2: There seems to be a gap between the control data and the start of the 127k simulation, is this a real data gap or an error in the figure?

This has now been moved to the Supplementary Material (SM2). However, to answer the reviewer: yes, there is a purposeful gap between the end of the control data and the start of the LIG simulation, because a number of model crashes caused the first ∼50 years of the spin-up to be unstable giving highly varied global mean temperatures. This is briefly noted in the figure caption (SM2)

Figure 2: Are the temperatures in the left-hand figure surface or 1.5 meter temperatures?

This has now been moved to the Supplementary Material (SM2). However, to answer the reviewer: these are 1.5m air temperatures, and this has now been clarified in the figure

Figure 2: Consider: 'b) TOA radiation balance'

This has now been moved to the Supplementary Material (SM2). However, to answer the reviewer: yes, this has now been corrected

Figure 2: This figure gives a good idea of the amount of internal variability in the system, which seems considerable in both the MH and lig127k simulations. Use this information to define which of your results are robust with respect to this internal variability. Is it true that variability is larger in the 'warm climates' than it is in de control?

We now use a Student's test (at the 99% level) as a matter of significant or robustness, which accounts for the interannual variability. Yes, it is true that variability is larger in the warm climate simulations than the PI, and this has been briefly noted in the text (lines 402)

Figure 3: For the control simulation the full depth is used instead of the top 1054 meters

according to the main text, please clarify.

This has now been corrected (see comment above)

Figure 4: These figures show some well-know climate change features, including polar amplification. The mechanisms of such spatial temperature anomaly patterns are not discussed. Provide a discussion or refer to previous work on the topic.

This now been addressed, with a short discussion on one of the mechanisms of polar amplification, namely sea-ice interactions, has now been added (lines 515-519), along with an accompanying figure in the Supplementary Material (SM6)

Figure 6: Rainfall anomalies on y-axis?

As part of other changes, this figure has now been removed. However, in subsequent zonal mean figures, the y-axis label has been changed from simply "Rainfall" to "Rainfall anomalies" (relative to the PI), which is what we understand the reviewer to mean here

Figure 6: x-axis values are not easy to read in this format.

As part of the changes, this figure has been removed

Figure 6: consider showing absolute precipitation values because I think those give a much better idea of the width of the wet and dry regions as discussed in the main text.

As part of the changes, this figure has been removed. However, when zonal mean rainfall is shown, both anomalies and absolute values are now shown (Figure 6)

Figure 6: Can't this figure be combined with figure 9?

This has now been done

Figure 8: Remove the ice core data points if the corresponding modeled surface temperature anomalies are not shown.

This has now been done

Figure 9: What does this figure add that is not already depicted in figures 10 and 11? Can't it be removed?

It was decided to remove existing figures 10 and 11 instead, because the same information is shown at the global scale in the new Figure 8 and Figure 10

Figure 12: why are the grey dashes that show required rainfall for grassland growth only start from 16 degrees north?

This has now been moved to the Supplementary Material (SM7), and it has been modified so that the model latitudes begin approximately where the rainfall threshold needed for grassland (the grey dashes) begins. To directly answer the reviewer, the grey dashes in this figure were taken directly from Figure 3a in Joussaume et al. (1999), which only has data beginning at 15.5°N. This reference has been added to the figure

Figure 12: Rainfall anomalies on y-axis?

This has now been corrected (see above comment)

Figure 12: Why are anomalies shown? Doesn't the threshold to support grassland depend on the absolute amount of precipitation?

Anomalies, rather than absolute values, are shown because this is following the methodology of Joussaume et al. (1999), who also considered annual mean anomalies in their study. Likewise, the region of averaging is larger here than Figure 6 (up to 30°E, as opposed to 15°E), again to be consistent with Joussaume et al. (1999). This has been made more clear in the text (lines 926)

TECHNICAL COMMENTS:

Line 41: are similar

This has been corrected

Line 41: period, but

This has been corrected

Line 53: generations of the same

As part of other changes, this sentence has now been removed

Line 121: therefore in the

This has been corrected

Line 146: consider removing "indeed"

This has been corrected

Line 149: double space before "Full"?

This has been corrected

Line 186: tuning of

This has been corrected

Line 201: including a reduction of the temperature bias in many regions

This has been corrected

Line 221: remove comma after 'design'

This has been corrected

Line 244: Too many brackets

This has been corrected

Line 272: annual mean surface

This has been corrected

Line 298: radiation balance?

As part of other changes, this sentence has now been removed

Line 384: in the core monsoon region?

As part of other changes, this has now been removed

Line 395: 'recent', what do you mean?

This has now been corrected

Line 399: what kind of uncertainty in simulated anomalies are you referring to, please clarify.

This has now been corrected

Line 400: remove 'often'

This has now been corrected

Line 437: small number of reconstructions?

As part of other changes, this has now been removed

Line 449: remove double comma

As part of other changes, this has now been removed

Line 499: refer to figure 10?

As part of other changes, this has now been removed

Line 505: smaller northward displacement?

As part of other changes, this has now been removed

Line 590: 'auspices', not sure if that is the right word for it.

This has now been corrected

Line 590: replace comma by a dot.

This has now been corrected

Line 591: remove 'time'?

This has now been corrected

Line 592: were assessed?

This has now been corrected

Line 603: 'time', are we talking seasons or different geological intervals?

This has now been corrected

Line 626: 'necessity' is perhaps a bit too strong in this context.

This has now been corrected

Line 1007: better not to use the & symbol.

This has now been corrected

Line 1008: for each

As part of other changes, this has now been removed

Line 1018: in this caption and some others the words 'simulated gridded anomalies' are used. This sounds a little double to me since nearly all climate models work on spatial grids so the output is per definition also gridded.

This has now been corrected

Lines 1024-1027: Is there no overlap between these two data sets? No single core was used in both of them?

As discussed in the Methodology section (section 2.3), the two datasets use different reference chronologies and methodologies to infer temporal surface temperature changes. Whilst they may use the same core, the methodologies are very different,

and therefore they should not be combined

Line 1033: erroneous bracket?

As part of other changes, this has now been removed

Please also note the supplement to this comment:
https://www.clim-past-discuss.net/cp-2019-160/cp-2019-160-AC1-supplement.pdf

---

## Author Comment (AC2) · 28 May 2020

To whom it may concern,

Re: Response letter to Reviewer 2, after submission of manuscript "The UK contribution to CMIP6/PMIP4: mid-Holocene and Last Interglacial experiments with HadGEM3, and comparison to the pre-industrial era and proxy data" by CJR Williams et al. to Climate of the Past (PMIP4 Special Edition).

I extend my sincere appreciation to the Reviewer 2 for their thorough examination of

my manuscript, and their detailed and highly constructive comments. I propose to address all of their concerns, both minor and major, so please see attached for a revised manuscript, still with the Track Changes included, to show my proposed modifications. I also attach a tidy version of this manuscript. Please note that the line numbers shown here relate to the Track Changes version.

Here, I address the reviewer's suggestions, comment-by-comment. I have also attached a PDF version of this letter, where the reviewer's comments are italicised and in a smaller font, and my corresponding response follows in a standard font.

I very much hope that my responses will satisfy the reviewer.

Yours faithfully,

Dr Charles JR Williams, and co-authors

—   REVIEWER 2

MAJOR COMMENTS:

1. It shall be elaborated what is new in this paper in terms of method, result and conclusion as compared to previous studies. Data model comparison in SST data and the question of seasonality could be more elaborated. It is not understandable that the SST comparison has not been performed on the MH experiment, although the data quality is higher and especially the dating uncertainty is much lower. Uncertainty is mentioned quite often, but not really elaborated. For the LIG, one could follow ideas outlined in Pfeiffer and Lohmann (CP) dealing with seasons and dating. For the MH, several data sets are available (e.g. Alkenone and Mg/Ca), again with uncertainties in the season or recorder depth.

Sentences have been added throughout the manuscript, including the abstract, introduction, methodology and conclusion, to elaborate and emphasise the novelty of this study. In short, we explain that although older versions of the UK model have been included in previous iterations of CMIP, and although present-day and future simulations from this model are included in CMIP6, this study is completely new because it is the first time this version of the model has been used to conduct any paleoclimate simulations. Give that these paleoclimate periods are out-of-sample in that they were not used in any way to tune or develop this model, these simulations provide a critical independent evaluation of the model's strengths and weaknesses.

Regarding the comment about including a model-data comparison with mid-Holocene SST data, we note that the manuscript already contains 5 separate datasets: i) land-surface temperature from the mid-Holocene, ii) land-surface precipitation from the mid-Holocene, iii) SST from the Last Interglacial (from 2 separate sources), and iv) precipitation from the Last Interglacial. The manuscript is already quite long, and we feel that the addition of more mid-Holocene SST data would not bring added information to the study. Moreover, another study (involving many of the co-authors here) is currently under review, looking specifically at the large-scale features during the mid-Holocene from CMIP6 models, including ours: Brierley et al. (2020). 'Large-scale features and evaluation of the PMIP4-CMIP6 midHolocene simulations'. Clim. Past. Under review. That study includes a significant model-data comparison section, including Holocene SSTs, and therefore we propose not to add any more model-data comparison in this paper, but rather to direct the reader on (lines 352-354).

2. The paper is too descriptive and focuses only on simulated temperature and precipitation. As a special contribution to CMIP6/PMIP4 is based on a single model, I would expect more comprehensive analysis, like the atmospheric and oceanic circulation, ocean states, and the potential relationship or mechanisms between different components. With such I believe the paper will meet the high standard of CP.

An example of atmospheric circulation changes (Figure 5) was already included in the original version of the manuscript, but this has now been elaborated in the text (lines 546-565). Moreover, in agreement with the reviewer, a measure of oceanic circulation has now been added (Figure 4), namely an example of the meridional overturning circulation. We find that there is almost no change in ocean circulation between the PI,

mid-Holocene and LIG simulations.

3. The authors show precipitation only for Africa. As a paper contributing to the CMIP6/PMIP4, it shall show the model behavior on global rather than regional scale.

This has now been done, showing both precipitation and temperature at both global and Africa-wide scales

MORE SPECIFIC COMMENTS:

1. Lines 94-104: This paragraph describes the previous studies on the modeled and observed MH and LIG states, which I find is too brief. As there are so many modelling studies and proxy papers, and this is directly linked to the present manuscript, thus I suggest to make more complete references. It is suggested to split the texts into two paragraphs, one describing the previous simulation results, the other the proxy issues.

This has now been addressed, with this section being expanded to include a number of other studies, as well as being divided into firstly a paragraph on the proxy data, and then a paragraph on modelling studies (lines 213-155)

2. Lines 106-108: The authors mention that the past warming are indeed different from future warming, as they are driven by quite different thermal forcing mechanisms, orbital parameters and greenhouse gases. I suggest to also mention that, i) the orbital forcing is shortwave and greenhouse gases are related to mainly the longwave radiation flux, ii) difference in orbital parameters leads to uneven horizontal and seasonal changes, but greenhouse gases can cause more uniform anomalies. Furthermore: iii) It is helpful to know the changes of greenhouse gases between MH/LIG and PI are equal to how much radiation flux anomalies? How to calculate such anomalies based on $CO_2$ changes can be found in some papers (e.g., Myhre, et al. 1998, GRL).

The first two elements have now been incorporated into the text (lines 163-165). Regarding the 3rd point, we feel that providing the precise calculation of the radiative forcing due to changes in MH and LIG greenhouse gases is beyond the scope of this study,

and would not provide a great deal of added information. However, this has been acknowledged and clarified in the text, with a reference to the Gunnar et al. (1998) study (lines 165-169).

3. Lines 161-203 Too detailed information in terms of the changes in model version is give here. I would recommend to simplify the text and to show what aspect/process can be improved in the newest model version. Details could be provided as supplementary material.

This has now been done, with much of the text being transferred to the Supplementary Material

4. Lines 205-209: The sensitivity and control experiments are performed on different platforms. I worry about how different the simulated climate can be. If possible, one shall show in the supplement the anomalies of surface temperature based on the same experiment

This issue is discussed in section 2.1.2 (lines 291-300), where a previous study (Guarino et al. 2020) compares simulations across different platforms and finds that the various climate variables discussed in this paper are not significantly different across platforms. Please see Figure 6 in Guarino et al. (2020) for an example of this.

5. Table 2 and Fig. 2a, the 1.5 m air temperature of LIG still show significant trend in the final years. Could you please show a trend map to check where such trend mainly occurs? Does it happen in the region of interest?

This has now been addressed, with a 1.5 m air temperature trend map for both climate simulations being shown in the supplementary material (SM3) and discussed in section 2.4 (lines 405-408)

6. Lines 297-323: I think it is not so necessary to describe the spin-up in such a detail. Just show the tables, and I also recommend to put Fig. 2 and Fig. 3 into the supplement.

As part of other changes, this has already been done (see above comments to Reviewer 1)

7. Fig. 4 and Fig. 5: Perform a Student's t-test to identify in which regions the anomalies are significant and which regions related to internal variabilities. Given the relatively short length of the MH and LIG experiments, it is very important to do so.

This has now been done to the new version of Figures 4 and 5

8. Line 334 'and'=>', and'

As part of other changes, this has now been removed

9. Lines 336-337: 'in order to' => 'to'

This has been corrected

10. Line 342: Title is confusing. The CMIP6 HadGEM3 simulations include the PI, right?

This has been corrected

11. Line 351: 'central' => 'Central'

This has been corrected

12. Line 359 and a lot of other places in the paper: please make the experiment name consistent throughout the paper, for example, use either MH or midHolocene, the same for LIG and lig127k, piControl and PI.

A new section detailing the terminology has now been added (section 2.1.1), to clarify exactly what term refers to either the simulations or the geological intervals (lines 216-226)

13. Line 371: greater land-sea contrast... Is it also the same case in your model? I would recommend to check the moist static energy instead of surface temperature, to also include the aspect of moisture.

As part of other changes, this has now been removed

14. Lines 374-377: the small anomalies... Again please use Student's t-test. Results discussed in the texts should have a significance level above 95%.

As above, this has now been included in the new versions of Figures 2 and 3, showing the 99% significant levels

15. Lines 373-374: Comparing Fig. 5a and 5b, I observe no obvious shift in ITCZ, only stronger monsoon rainfall in LIG compared to MH.

As part of other changes, this sentence has now been modified and clarified

16. caption of Fig. 6, 9, and 12: Generally West Africa should be within 20W-15E. Why take 20W-30E?

This has now been corrected, such that all zonal mean plots go from 20°W-15°E

17. Lines 398-400: Please explain where the large uncertainty in proxy comes from.

This has now been corrected

18. Lines 422-424: Can this underestimation of the warming be used to explain the "Holocene temperature conundrum"? Or, might the "Holocene temperature conundrum" be caused by the fact that most of the proxy locate in regions with positive temperature anomalies? The proxy data represent seasonal or annual mean value? It might be helpful to discuss these issues. See, e.g. Lohmann et al. (2013, CP) for a comprehensive comparison for SST changes during the Holocene.

We agree with the reviewer that this term is ambiguous, and it has therefore been removed

19. Line 396: It would be better to clarify here the threshold of RMSE (is there any?) for a reasonable simulation result, in terms of surface temperature, precipitation and sst.

We do not use a threshold of RMSE, but we have clarified this in the text (lines 614)

20. Line 447: if => but

As part of other changes, this has now been removed

21. Fig. 10 and 11: Again, please show significance (t-test).

As part of other changes, these figures have now been removed

22. Line 557. The model used prescribed vegetation, and does not consider dust. Please discuss the influence of the lack of interactive vegetation and dust on the Africa monsoon rainfall.

This has now been addressed (lines 933-935)

23. Optional: I encourage the author to make a separate discussion section.

As part of other changes, the summary and conclusions have now been restructured and rewritten, and further discussion has been added throughout the results section

Please also note the supplement to this comment:
https://www.clim-past-discuss.net/cp-2019-160/cp-2019-160-AC2-supplement.pdf

---

## Editor Decision (ED1)

Dear authors,

I am happy with the way you reorganized your manuscript to take into account the reviewers' comments. The manuscript is now almost ready for publication.

However, I still have comments.

1.  Minor comment

    I invite you to double check the writing of the supplementary material and make it 'easier' to read. In particular the first sentence (l21-22) looks very strange to me. I think it would be worth to remind the readers what are GA, GC, GO, GIS (only the 'full words' for the acronyms, such as 'Global Atmosphere', 'Global Ocean', … nothing more. And would it be possible to explain in a few words what you mean with 'bottom-up and top-down developments' (l24-35)?

2.  Major comment

    Your data availability section does not comply with the requirements from COPERNICUS. I asked the Climate-of-Past co-editors-in-chief and I copy here the answer:

    *The data availability is where the authors indicate the DOIs of the datasets.*

    *Now I checked the indicated datasets and some of datasets in this paper are not freely accessible. In such case I wish that the dataset is available elsewhere in PANGAEA or NOAA.*

    *The authors indicate that they WILL be uploaded to the "ESGF …", this site should have an internet address that could be available and therefore mentioned. I am personally against papers indicating what the authors are presently indicating that something should happen sometimes because this will remain like that unless some corrigendum is published.*

    Here are two examples of how the data should be cited:

    *datasets may be listed in the data availability with a reference like "Martrat et al. (2004b)" which in fact refers to the following reference in the bibliography list: "Martrat, B., Grimalt, J. O., Shackleton, N. J., de Abreu, L., Hutterli, M. A., and Stocker, T. F.: Sea surface temperature estimation for the Iberian Margin, PANGAEA, [https://doi.org/10.1594/PANGAEA.771894](https://doi.org/10.1594/PANGAEA.771894), 2007b. Therefore in theory you get 2 citations for the same work, one for the published paper and another one for the deposited dataset.*

    *Now for the NCDC-NOAA dataset, there is no DOI. Therefore no citation of the dataset in the reference list but the citation in the data availability section is ["https://www.ncdc.noaa.gov/paleo-search/study/20127 (last access: 8 December 2018, Mokeddem and McManus, 2016)"](https://www.ncdc.noaa.gov/paleo-search/study/20127)*

    *Sorry to be a bit technical but this the new policy and publishing process related to FAIR.*

I am looking forward to reading your amended data availability section,

Marie-France Loutre

---

## Author Response (AR2)

**Dr Charles JR Williams, BA DPhil FRGS**
**Research Fellow**
1.2n, School of Geographical Sciences
University Road, Bristol, BS8 1SS

Wednesday 10 June 2020

Dear Dr Loutre,

**Re: Submission of manuscript "*CMIP6/PMIP4 simulations of the mid-Holocene and Last Interglacial using HadGEM3: comparison to the pre-industrial era, previous model versions, and proxy data*" by CJR Williams *et al*. to Climate of the Past (PMIP4 Special Edition).**

Thank you very much for your most recent comments. I have now made all the corrections that you requested, so please see below for my responses:

*1. Minor comment*

*I invite you to double check the writing of the supplementary material and make it 'easier' to read. In particular the first sentence (l21-22) looks very strange to me. I think it would be worth to remind the readers what are GA, GC, GO, GIS (only the 'full words' for the acronyms, such as 'Global Atmosphere', 'Global Ocean', … nothing more. And would it be possible to explain in a few words what you mean with 'bottom-up and top-down developments' (l24-35)?*

This has now been corrected, with the first sentence being changed so that it makes more logical sense (and is more independent from the main manuscript), the acronyms being listed, and that last sentence being removed (as we considered it ambiguous)

2. Major comment

Your data availability section does not comply with the requirements from COPERNICUS. I asked the Climate-of-Past co-editors-in-chief and I copy here the answer…

As requested, I have now added in the website to the ESGF portal, and have made it clearer that although my simulations have not yet been uploaded to this, they are nevertheless publicly available, by contacting myself. As I say in my manuscript, we plan to upload our simulations to the ESGF portal, and will do so in the near future. However, CMIP6 protocol states that the data must all be in a correct and consistent format - a process called CMORising - and this is a lengthy and nontrivial process. It would seriously delay publication of this manuscript if the data have to be uploaded to the ESGF before publication. I hope the clarification that I have added will satisfy your regulations? If not, would an acceptable alternative be that I provide climatologies of the relevant fields (temperature, precipitation etc) from the model simulations as a supplementary netcdf file? Please let me know the best option.

Lastly, I have also corrected the reference list, as per your comment in the email dated 2 June 2020:

Please note that your reference list has not been compiled according to our standards. Please consider adjusting your reference list with the next revision of your manuscript. The manuscript preparation guidelines can be seen at: https://www.climate-of-the-past.net/for_authors/manuscript_preparation.html.

This has now been corrected.

I very much hope that my manuscript now meets your technical specifications, and is deemed acceptable and ready for publication.

Yours sincerely,

Dr Charles JR Williams, and co-authors

[revised manuscript text omitted]
̶c̶o̶m̶p̶a̶r̶i̶n̶g̶ ̶t̶h̶e̶ ̶r̶e̶s̶u̶l̶t̶s̶ ̶w̶i̶t̶h̶ ̶a̶v̶a̶i̶l̶a̶b̶l̶e̶ ̶p̶r̶o̶x̶y̶ ̶d̶a̶t̶a̶,̶

̶p̶r̶e̶v̶i̶o̶u̶s̶ ̶v̶e̶r̶s̶i̶o̶n̶s̶ ̶o̶f̶ ̶t̶h̶e̶ ̶U̶K̶'̶s̶ ̶s̶a̶m̶e̶ ̶p̶h̶y̶s̶i̶c̶a̶l̶ ̶c̶l̶i̶m̶a̶t̶e̶ ̶m̶o̶d̶e̶l̶,̶ ̶a̶n̶d̶ ̶o̶t̶h̶e̶r̶ ̶m̶o̶d̶e̶l̶s̶ ̶f̶r̶o̶m̶ ̶C̶M̶I̶P̶5̶.̶  In addition to a global assessment, a secondary T̶h̶e̶ focus of this paper is on the fidelity of the temperature anomalies g̶l̶o̶b̶a̶l̶l̶y̶ and the degree of precipitation enhancement in the Sahara, the latter of which has proved problematic for several generations of models.  T̶h̶e̶ ̶r̶e̶s̶u̶l̶t̶s̶ ̶d̶i̶s̶c̶u̶s̶s̶e̶d̶ ̶h̶e̶r̶e̶ ̶a̶r̶e̶

̶s̶p̶l̶i̶t̶ ̶i̶n̶t̶o̶ ̶t̶w̶o̶ ̶s̶e̶c̶t̶i̶o̶n̶s̶:̶ ̶a̶f̶t̶e̶r̶ ̶a̶n̶ ̶a̶s̶s̶e̶s̶s̶m̶e̶n̶t̶ ̶o̶f̶ ̶t̶h̶e̶ ̶l̶e̶v̶e̶l̶ ̶o̶f̶ ̶e̶q̶u̶i̶l̶i̶b̶r̶i̶u̶m̶ ̶g̶a̶i̶n̶e̶d̶ ̶d̶u̶r̶i̶n̶g̶ ̶t̶h̶e̶ ̶s̶p̶i̶n̶-̶u̶p̶ ̶p̶h̶a̶s̶e̶,̶

̶t̶h̶e̶ ̶m̶a̶i̶n̶ ̶f̶o̶c̶u̶s̶ ̶i̶s̶ ̶o̶n̶ ̶t̶h̶e̶ ̶m̶o̶d̶e̶l̶-̶d̶a̶t̶a̶ ̶a̶n̶d̶ ̶m̶o̶d̶e̶l̶-̶m̶o̶d̶e̶l̶ ̶c̶o̶m̶p̶a̶r̶i̶s̶o̶n̶s̶ ̶u̶s̶i̶n̶g̶ ̶t̶h̶e̶ ̶p̶r̶o̶d̶u̶c̶t̶i̶o̶n̶ ̶r̶u̶n̶s̶.̶

Following this introduction, Section 2 describes the model, the experimental design, a̶n̶d̶ the proxy data used for the model-data comparisons, and a brief discussion of the simulation spin-up phases.

Section 3 then presents the results, beginning with the model-model comparison and following with the model-data comparison, and  ̶f̶d̶i̶v̶i̶d̶e̶d̶ ̶i̶n̶t̶o̶ ̶t̶w̶o̶ ̶s̶u̶b̶s̶e̶c̶t̶i̶o̶n̶s̶:̶ ̶i̶)̶ ̶e̶q̶u̶i̶l̶i̶b̶r̶i̶u̶m̶ ̶d̶u̶r̶i̶n̶g̶ ̶t̶h̶e̶ ̶s̶p̶i̶n̶-̶u̶p̶

̶p̶h̶a̶s̶e̶;̶ ̶a̶n̶d̶ ̶i̶i̶)̶ ̶m̶o̶d̶e̶l̶-̶d̶a̶t̶a̶ ̶a̶n̶d̶ ̶m̶o̶d̶e̶l̶-̶m̶o̶d̶e̶l̶ ̶c̶o̶m̶p̶a̶r̶i̶s̶o̶n̶s̶ ̶f̶r̶o̶m̶ ̶t̶h̶e̶ ̶p̶r̶o̶d̶u̶c̶t̶i̶o̶n̶ ̶r̶u̶n̶s̶.̶  F̶i̶n̶a̶l̶l̶y̶finally, section 4 summarises and concludes.

**2. MODEL, EXPERIMENT DESIGN, A̶N̶D̶ DATA AND SPIN-UP SIMULATIONS**

**2.1. Model**

*2.1.1. Model terminology*

In this paper, and consistent with CMIP nomenclature, the 'spin-up phase' of the simulations refers to when they are spinning up to atmospheric and oceanic equilibrium, whereas the 'production run'

refers to the end parts (usually the last 50 or 100 years) of the simulation used to calculate the climatologies, presented as the results.  When discussed as geological intervals, the preindustrial, mid-

Holocene and Last Interglacial are referred to as the PI, MH and LIG respectively.  In contrast, when discussed as the three most recent simulations using HadGEM3 (see below), consistent with CMIP

they are referred to as the *piControl*, *midHolocene* and *lig127k* simulations, respectively.  When the

*midHolocene* and *lig127k* are discussed collectively, they are referred to as the 'warm climate simulations'; whilst it is acknowledged that other factors differentiate these simulations such as orbital configuration or $CO_2$, 'warm climate simulations' was deemed an appropriate collective noun.

*2.1.2. Model details*

The M̶H̶ ̶a̶n̶d̶ ̶L̶I̶G̶ ̶s̶i̶m̶u̶l̶a̶t̶i̶o̶n̶s̶ ̶c̶o̶n̶d̶u̶c̶t̶e̶d̶ ̶h̶e̶r̶e̶ ̶(̶r̶e̶f̶e̶r̶r̶e̶d̶ ̶t̶o̶ ̶a̶s̶ *m̶i̶d̶H̶o̶l̶o̶c̶e̶n̶e̶* ̶a̶n̶d̶ *l̶i̶g̶1̶2̶7̶k̶*,̶ ̶r̶e̶s̶p̶e̶c̶t̶i̶v̶e̶l̶y̶,̶

̶a̶n̶d̶ ̶c̶o̶l̶l̶e̶c̶t̶i̶v̶e̶l̶y̶ ̶a̶s̶ ̶t̶h̶e̶ warm climate simulations conducted here), and i̶n̶d̶e̶e̶d̶ the *piControl* P̶I̶

[revised manuscript text omitted]

**SUPPLEMENTARY MATERIAL**

**TEXT**

**Model description**

Here, a  description of the major changes in the latest version of UK's physical climate model, HadGEM3-GC3.1 , relative to its predecessor,  is given, following on directly from

Section 2.1.2 in the main manuscript . Beginning with Global Atmosphere (GA7) and

Global Land components (GL7), a once-in-a-decade replacement of the model's dynamical core, implementing ENDGame, was undertaken for the previous version (GA6) and therefore remains the same in GA7 (Walters *et al*. 2017)[1]. A number of other developments, since the previous version of the model, have also been included.

[revised manuscript text omitted]

---

## Author Response (AR3)

**Dr Charles JR Williams, BA DPhil FRGS**
**Research Fellow**
1.2n, School of Geographical Sciences
University Road, Bristol, BS8 1SS

Friday 3 July 2020

Dear Dr Loutre,

**Re: Submission of manuscript "*CMIP6/PMIP4 simulations of the mid-Holocene and Last Interglacial using HadGEM3: comparison to the pre-industrial era, previous model versions, and proxy data*" by CJR Williams *et al*. to Climate of the Past (PMIP4 Special Edition).**

Thank you very much for your most recent comments and emails. I have now made the necessary changes to the Data Availability section, consistent with the Copernicus editorial instructions, and have included as Supplementary Material the data going into the manuscript figures.

I very much hope that my manuscript now meets your technical specifications, and is deemed acceptable and ready for publication.

Yours sincerely,

Dr Charles JR Williams, and co-authors